# Causal Temporal Representation Learning with Nonstationary Sparse Transition

**Xiangchen Song**[1]    **Zijian Li**[2]    **Guangyi Chen**[1,2]    **Yujia Zheng**[1]
**Yewen Fan**[1]    **Xinshuai Dong**[1]    **Kun Zhang**[1,2]
[1]Carnegie Mellon University
[2]Mohamed bin Zayed University of Artificial Intelligence
{xiangchensong,kunz1}@cmu.edu

## Abstract

Causal Temporal Representation Learning (Ctrl) methods aim to identify the temporal causal dynamics of complex nonstationary temporal sequences. Despite the success of existing Ctrl methods, they require either directly observing the domain variables or assuming a Markov prior on them. Such requirements limit the application of these methods in real-world scenarios when we do not have such prior knowledge of the domain variables. To address this problem, this work adopts a sparse transition assumption, aligned with intuitive human understanding, and presents identifiability results from a theoretical perspective. In particular, we explore under what conditions on the significance of the variability of the transitions we can build a model to identify the distribution shifts. Based on the theoretical result, we introduce a novel framework, *Causal Temporal Representation Learning with Nonstationary Sparse Transition* (**CtrlNS**), designed to leverage the constraints on transition sparsity and conditional independence to reliably identify both distribution shifts and latent factors. Our experimental evaluations on synthetic and real-world datasets demonstrate significant improvements over existing baselines, highlighting the effectiveness of our approach.

## 1 Introduction

Causal learning from sequential data remains a fundamental yet challenging task [1–3]. Discovering temporal causal relations among *observed* variables has been extensively studied in the literature [4–6]. However, in many real-world scenarios such as video understanding [7], observed data are generated by causally related latent temporal processes or confounders rather than direct causal edges. This leads to the task of *causal temporal representation learning* (Ctrl), which aims to build compact representations that concisely capture the data generation processes by inverting the mixing function that transforms latent factors into observations and identifying the transitions that govern the underlying latent causal dynamics. This learning problem is known to be challenging without specific assumptions [8, 9]. The task becomes significantly more complex with *nonstationary* transitions, which are often characterized by multiple distribution shifts across different domains, particularly when these domains or shifts are also unobserved.

Recent advances in unsupervised representation learning, particularly through nonlinear Independent Component Analysis (ICA), have shown promising results in identifying latent variables by incorporating side information such as class labels and domain indices [10–19]. For time-series data, historical information is widely utilized to enhance the identifiability of latent temporal causal processes [20–23]. However, existing studies primarily derive results under stationary conditions [11, 21] or nonstationary conditions with observed domain indices [13, 22, 23]. These methods are limited in application as general time series data are typically nonstationary and domain information is difficult

to obtain. Recent studies [15, 24–26] have adopted a Markov structure to handle nonstationary domain variables and can infer domain indices directly from observed data. (More related work can be found in Appendix S4.) However, these methods face significant limitations; some are inadequate for modeling time-delayed causal relationships in latent spaces, and they rely on the Markov property, which cannot adequately capture the arbitrary nonstationary variations in domain variables. This leads us to the following important yet unresolved question:

*How can we establish identifiability of nonstationary nonlinear ICA for general sequence data without knowledge of the prior distribution of the domain variables?*

Relying on observing domain variables or known Markov priors to capture nonstationarity seems counter-intuitive, especially considering how easily humans can identify domain shifts given sufficient variation on transitions, such as video action segmentation [27, 28] and recognition [29–31] tasks. In this work, we theoretically investigate the conditions on the significance of transition variability to identify distribution shifts. The core idea is transition clustering, assuming transitions within the same domain are similar, while transitions across different domains are distinct. Building on this identification theorem, we propose *Causal Temporal Representation Learning with Nonstationary Sparse Transition* (**CtrlNS**), to identify both distribution shifts and latent temporal dynamics. Specifically, we constrain the complexity of the transition function to identify domain shifts. Subsequently, with the identified domain variables, we learn the latent variables using conditional independence constraints. These two processes are jointly optimized within a VAE framework.

The main contributions of this work are as follows: (1) To our best knowledge, this is the first identifiability result that handles nonstationary time-delayed causally related latent temporal processes without knowledge of the prior distribution of the domain variables. (2) We present **CtrlNS**, a principled VAE-based framework for recovering both nonstationary domain variables and time-delayed latent causal dynamics. (3) Experiments on synthetic and real-world datasets demonstrate the effectiveness of the proposed method in recovering latent variables and domain indices.

## 2 Problem Formulation

### 2.1 Nonstationary Time Series Generative Model

We first introduce a nonstationary time-series generative model in our setting. The observational dataset is $\mathcal{D} = \{\mathbf{x}_t\}_{t=1}^T$, where $\mathbf{x}_t \in \mathbb{R}^n$ is produced from causally related, time-delayed latent components $\mathbf{z}_t \in \mathbb{R}^n$ through an invertible mixing function $\mathbf{g}$:

$$\mathbf{x}_t = \mathbf{g}(\mathbf{z}_t). \tag{1}$$

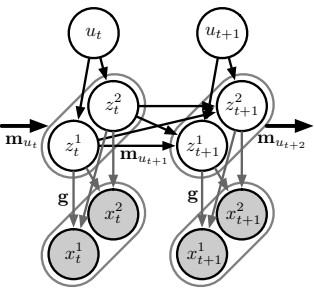

In the nonstationary setting, transitions within the latent space vary over time. Define $u$ as the domain or regime index variable, with $u_t$ corresponding to domain variable at time step $t$. Assuming there are $U$ distinct regimes, i.e., $u_t \in \{1, 2, \ldots, U\}$, each regime exhibits unknown distribution shifts. Those regimes are characterized by $U$ different transition functions $\{\mathbf{m}_u\}_{u=1}^U$, which were originally introduced in [32] through change factors to capture these distribution shifts in transition dynamics. The $i$-th component of latent variable $\mathbf{z}_t$, is then generated via $i$-th component of transition function $\mathbf{m}$:

Figure 1: Graphical model for nonstationary causally related time-delayed time-series data generation process with unobserved domain variables $u_t$.

$$z_{t,i} = m_i \left( u_t, \{z_{t',j} \mid z_{t',j} \in \mathbf{Pa}(z_{t,i})\}, \epsilon_{t,i} \right), \tag{2}$$

where $\mathbf{Pa}(z_{t,i})$ represents the set of latent factors directly influencing $z_{t,i}$, which may include any subset of $\mathbf{z}_{<t} = \{z_{\tau,i} \mid \tau \in \{1, 2, \ldots, t-1\}, i \in \{1, 2, \ldots, n\}\}$. For analytical simplicity, we assume that the parents in the causal graph are restricted to elements in $\mathbf{z}_{t-1}$. Extensions to higher-order cases, which involve multistep, time-delayed causal relations, are already discussed in Appendix S1.5 of [23]. These extensions are orthogonal to our contributions and are therefore omitted here for brevity. Importantly, in a nonstationary context, $\mathbf{Pa}(\cdot)$ may also be sensitive to the domain index $u_t$, indicating that causal dependency graphs vary across different domains or regimes, which will be revisited in our later discussion on identifiability. We assume that the generation processes for each $i$-th component of $\mathbf{z}_t$ are mutually independent, given $\mathbf{z}_{<t}$ and $u_t$. Consistent with the existing

literature [23, 25], we further assume that the noise terms $\epsilon_{t,i}$ are independent both spatially and temporally. This assumption implies that there is no instantaneous causal influence among the latent causal processes. The graphical model corresponding to this setting is illustrated in Figure 1.

## 2.2 Identifiability of Domain Variables and Latent Causal Processes

In this section we introduce the identifiability of both domain variables and time-delayed latent causal processes in Definitions 2 and 3, respectively. If the estimated latent processes are identifiable at least up to a permutation and component-wise invertible transformations, then the latent causal relationships are also immediately identifiable. This follows from the fact that conditional independence relations comprehensively characterize the time-delayed causal relations within a time-delayed causally sufficient system, in which there are no latent causal confounders in the causal processes. Notably, invertible component-wise transformations on latent causal processes preserve their conditional independence relationships. We now present definitions concerning observational equivalence, the identifiability of domain variables and latent causal processes.

**Definition 1** (Observational Equivalence). *Formally, consider $\{\mathbf{x}_t\}_{t=1}^T$ as a sequence of observed variables generated by true temporally causal latent processes specified by $(\mathbf{m}, \mathbf{u}, p(\epsilon), \mathbf{g})$ given in Eqs. (1) and (2). Here, $\mathbf{m}$ and $\epsilon$ denote the concatenated vector form across $n$ dimensions in the latent space. Similarly $\mathbf{u}$ for time steps $1$ to $T$. A learned generative model $(\hat{\mathbf{m}}, \hat{\mathbf{u}}, \hat{p}(\epsilon), \hat{\mathbf{g}})$ is observationally equivalent to the ground truth one $(\mathbf{m}, \mathbf{u}, p(\epsilon), \mathbf{g})$ if the model distribution $p_{\hat{\mathbf{m}},\hat{\mathbf{u}},\hat{p}_\epsilon,\hat{\mathbf{g}}}(\{\mathbf{x}_t\}_{t=1}^T)$ matches the data distribution $p_{\mathbf{m},\mathbf{u},p_\epsilon,\mathbf{g}}(\{\mathbf{x}_t\}_{t=1}^T)$ everywhere.*

**Definition 2** (Identifiable Domain Variables). *Domain variables are said to be identifiable up to label swapping if observational equivalence (Def. 1) implies identifiability of domain variables up to a permutation $\sigma$ for domain indices:*

$$p_{\hat{\mathbf{m}},\hat{\mathbf{u}},\hat{p}_\epsilon,\hat{\mathbf{g}}}(\{\mathbf{x}_t\}_{t=1}^T) = p_{\mathbf{m},\mathbf{u},p_\epsilon,\mathbf{g}}(\{\mathbf{x}_t\}_{t=1}^T) \Rightarrow \hat{u}_t = \sigma(u_t), \forall t \in \{1, 2, \ldots, T\}. \quad (3)$$

**Definition 3** (Identifiable Latent Causal Processes). *The latent causal processes are said to be identifiable if observational equivalence (Def. 1) leads to the identifiability of latent variables up to a permutation $\pi$ and component-wise invertible transformation $\mathcal{T}$:*

$$p_{\hat{\mathbf{m}},\hat{\mathbf{u}},\hat{p}_\epsilon,\hat{\mathbf{g}}}(\{\mathbf{x}_t\}_{t=1}^T) = p_{\mathbf{m},\mathbf{u},p_\epsilon,\mathbf{g}}(\{\mathbf{x}_t\}_{t=1}^T) \Rightarrow \hat{\mathbf{g}}^{-1}(\mathbf{x}_t) = \mathcal{T} \circ \pi \circ \mathbf{g}^{-1}(\mathbf{x}_t), \quad \forall \mathbf{x}_t \in \mathcal{X}, \quad (4)$$

*where $\mathcal{X}$ denotes the observation space.*

# 3 Identifiability Theory

In this section, we demonstrate that under mild conditions, the domain variables $u_t$ are identifiable up to label swapping and the latent variables $\mathbf{z}_t$ are identifiable up to permutation and component-wise transformations. We partition our theoretical discussion into two sections: (1) identifiability of nonstationary discrete domain variables $u_t$ and (2) identifiability of latent causal processes. We slightly extend the usage of $\mathrm{supp}(\cdot)$ to define the square matrix support and the support of a square matrix function as follows:

**Definition 4** (Matrix Support). *The support (set) of a square matrix $\mathbf{A} \in \mathbb{R}^{n \times n}$ is defined using the indices of non-zero entries as:*

$$\mathrm{supp}(\mathbf{A}) \coloneqq \{(i,j) \mid \mathbf{A}_{i,j} \neq 0\}. \quad (5)$$

**Definition 5** (Matrix Function Support). *The support (set) of a square matrix function $\mathbf{A} : \Theta \to \mathbb{R}^{n \times n}$ is defined as:*

$$\mathrm{supp}(\mathbf{A}(\Theta)) \coloneqq \{(i,j) \mid \exists \theta \in \Theta, \mathbf{A}(\theta)_{i,j} \neq 0\}. \quad (6)$$

For brevity, let $\mathcal{M}$ and $\widehat{\mathcal{M}}$ denote the $n \times n$ binary matrices representing the support of the Jacobian $\mathbf{J}_{\mathbf{m}}(\mathbf{z}_t)$ and $\mathbf{J}_{\hat{\mathbf{m}}}(\hat{\mathbf{z}}_t)$, respectively. The $(i,j)$-th entry of $\mathcal{M}$ is 1 if and only if $(i,j) \in \mathrm{supp}(\mathbf{J}_{\mathbf{m}})$. We further define the transition complexity using its Fréchet norm as $|\mathcal{M}| = \sum_{i,j} \mathcal{M}_{i,j}$, and similarly for $\widehat{\mathcal{M}}$. In the nonstationary setting, this support matrix becomes a function of the domain index $u$, denoted as $\mathcal{M}_u$ and $\widehat{\mathcal{M}}_u$. Additionally, we introduce the concept of weakly diverse lossy transitions for the data generation process, which is formally defined below:

**Definition 6** (Weakly Diverse Lossy Transition). *The set of transition functions described in Eq.* (2) *is said to be diverse lossy if it satisfies the following conditions:*

1. *(Lossy) For every time and indices tuple $(t, i, j)$ with edge $z_{t-1,i} \rightarrow z_{t,j}$ representing a causal link defined with the parents set $\mathbf{Pa}(z_{t,j})$ in Eq. 2, transition function $m_j$ is a lossy transformation w.r.t. $z_{t-1,i}$ i.e., there exists an open set $S_{t,i,j}$[1], changing $z_{t-1,i}$ within this set will not change the value of $m_j$, i.e. $\forall z_{t-1,i} \in S_{t,i,j}$, $\frac{\partial m_j}{\partial z_{t-1,i}} = 0$.*

2. *(Weakly Diverse) For every element $z_{t-1,i}$ of the latent variable $\mathbf{z}_{t-1}$ and its corresponding children set $\mathcal{J}_{t,i} = \{j \mid z_{t-1,i} \in \mathbf{Pa}(z_{t,j}), j \in \{1, 2, \dots, n\}\}$, transition functions $\{m_j\}_{j \in \mathcal{J}_{t,i}}$ are weakly diverse i.e., the intersection of the sets $S_{t,i} = \cap_{j \in \mathcal{J}_{t,i}} S_{t,i,j}$ is not empty, and such sets are diverse, i.e., $S_{t,i} \neq \emptyset$, and $S_{t,i,j} \setminus S_{t,i} \neq \emptyset, \forall j \in \mathcal{J}_{t,i}$.*

### 3.1 Identifiability of Domain Variables

**Theorem 1** (Identifiability of Domain Variables). *Suppose that the dataset $\mathcal{D}$ are generated from the nonstationary data generation process as described in Eqs.* (1) *and* (2)*. Suppose the transitions are weakly diverse lossy (Def. 6) and the following assumptions hold:*

i. *(Mechanism Separability) There exists a ground truth mapping $\mathcal{C} : \mathcal{X} \times \mathcal{X} \rightarrow \mathcal{U}$ determined the real domain indices, i.e., $u_t = \mathcal{C}(\mathbf{x}_{t-1}, \mathbf{x}_t)$.*

ii. *(Mechanism Sparsity) The estimated transition complexity on dataset $\mathcal{D}$ is less than or equal to ground truth transition complexity, i.e., $\mathbb{E}_{\mathcal{D}}|\widehat{\mathcal{M}}_{\hat{u}}| \leq \mathbb{E}_{\mathcal{D}}|\mathcal{M}_u|$.*

iii. *(Mechanism Variability) Mechanisms are sufficiently different. For all $u \neq u'$, $\mathcal{M}_u \neq \mathcal{M}_{u'}$ i.e. there exists index $(i, j)$ such that $[\mathcal{M}_u]_{i,j} \neq [\mathcal{M}_{u'}]_{i,j}$.*

*Then the domain variables $u_t$ is identifiable up to label swapping (Def. 2).*

Theorem 1 states that if we successfully learn a set of estimated transitions $\{\hat{\mathbf{m}}_u\}_{u=1}^{U}$, the decoder $\hat{\mathbf{g}}$, and the domain clustering assignment $\hat{\mathcal{C}}$, where $\hat{\mathbf{m}}_u$ corresponds to the estimation of Eq. (2) for a particular regime or domain $u$, and the system can fit the data as follows:

$$\hat{\mathbf{x}}_t = \hat{\mathbf{g}} \circ \hat{\mathbf{m}}_{\hat{u}_t} \circ \hat{\mathbf{g}}^{-1}(\mathbf{x}_{t-1}) \quad \text{and} \quad \hat{u}_t = \hat{\mathcal{C}}(\mathbf{x}_{t-1}, \mathbf{x}_t), \tag{7}$$

assuming that the transition complexity is kept low (as per Assumption ii). Then the estimated domain variables $\hat{u}_t$ must be the true domain variables $u_t$ up to a permutation.

**Proof sketch** The core idea of this proof is to demonstrate that the global minimum of transition complexity can only be achieved when the domain variables $u_t$ are correctly estimated. (1) First, we consider the case when we have an optimal decoder estimation $\hat{\mathbf{g}}^*$ which is a component-wise transformation of the ground truth, incorrect estimations of $u_t$ will strictly increase the transition complexity, i.e., $\mathbb{E}_{\mathcal{D}}|\widehat{\mathcal{M}}_{\hat{u}}^*| > \mathbb{E}_{\mathcal{D}}|\widehat{\mathcal{M}}_u^*|$. (2) Second, we show that with arbitrary estimations $\hat{u}_t$, the transition complexity for any non-optimal decoder estimation $\hat{\mathbf{g}}$ will be equal to or higher than that for the optimal $\hat{\mathbf{g}}^*$, i.e., $\mathbb{E}_{\mathcal{D}}|\widehat{\mathcal{M}}_{\hat{u}}| \geq \mathbb{E}_{\mathcal{D}}|\widehat{\mathcal{M}}_{\hat{u}}^*|$. Thus, the global minimum of transition complexity can only be achieved when $u_t$ is optimally estimated, which must be a permuted version of the ground truth domain variables $u_t$. A comprehensive proof can be found in Appendix S1.1.

### 3.2 Remark on Mechanism Variability

The assumption of mechanism variability, as stated in Assumption iii, requires that the Jacobian support matrices differ across domains, indicating that the causal graph connecting past states ($\mathbf{z}_{t-1}$) to current states ($\mathbf{z}_t$) must differ by at least one edge. Addressing scenarios where the causal graphs remain identical but the transition functions associated with the edges vary is generally challenging without imposing additional assumptions. A more detailed discussion of the difficulties involved in such cases is provided in Appendix S1.4.4. To effectively address these scenarios, we extend the concept of the Jacobian support matrix by incorporating higher-order derivatives. This extension

---

[1]We implicitly assume $S_{t,i,j}$ together with $S_{t,i}$, $S_{t,i,j} \setminus S_{t,i}$ have non-zero measure.

provides a more detailed characterization of the variability in transition functions across different domains. We now present the following definition to formalize this concept:

**Definition 7** (Higher Order Partial Derivative Support Matrix). *The $k$-th order partial derivative support matrix for transition $\mathbf{m}$ denoted as $\mathcal{M}^k$ is a binary $n \times n$ matrix with*

$$\left[\mathcal{M}^k\right]_{i,j} = 1 \iff \exists \mathbf{z} \in \mathcal{Z}, \frac{\partial^k m_j}{\partial z_i^k} \neq 0. \tag{8}$$

We utilize the variability in the higher-order partial derivative support matrix to extend the identifiability results of Theorem 1. This extension applies to cases where the causal graphs remain identical across two domains, yet the transition functions take different forms.

**Corollary 1** (Identifiability under Function Variability). *Suppose the data $\mathcal{D}$ is generated from the nonstationary data generation process described in (1) and (2). Assume the transitions are weakly diverse lossy (Def. 6), and the mechanism separability assumption i along with the following assumptions hold:*

    *v. (Mechanism Function Variability) Mechanism Functions are sufficiently different. There exists $K \in \mathbb{N}$ such that for all $u \neq u'$, there exists $k \leq K$, $\mathcal{M}_u^k \neq \mathcal{M}_{u'}^k$, i.e. there exists index $(i,j)$ such that $\left[\mathcal{M}_u^k\right]_{i,j} \neq \left[\mathcal{M}_{u'}^k\right]_{i,j}$.*

    *vi. (Higher Order Mechanism Sparsity) The estimated transition complexity on dataset $\mathcal{D}$ is no more than ground truth transition complexity,*

$$\mathbb{E}_{\mathcal{D}} \sum_{k=1}^{K} |\widehat{\mathcal{M}}_{\hat{u}}^k| \leq \mathbb{E}_{\mathcal{D}} \sum_{k=1}^{K} |\mathcal{M}_u^k|. \tag{9}$$

*Then the domain variables $u_t$ are identifiable up to label swapping (Def. 2).*

To prove this corollary, we leverage the property that, for any two distinct domains, there exists an edge in the causal graph such that the supports of their $k$-th order partial derivatives differ. This difference ensures the separability of the two domains. A detailed proof can be found in Appendix S1.2.

### 3.3 Identifiability of Latent Causal Process

Once the identifiability of $u_t$ is achieved, the problem reduces to a nonstationary temporal nonlinear ICA with *observed* domain index. Leveraging the sufficient variability approach proposed in [23], we demonstrate full identifiability of the data generation process. This sufficient variability concept is further incorporated into the following lemma, adapted from Theorem 2 in [23]:

**Lemma 1** (Theorem 2 in Yao et al., [23]). *Suppose that the data $\mathcal{D}$ are generated from the nonstationary data generation process as described in Eqs. (1) and (2). Let $\eta_{kt}(u)$ denote the logarithmic density of $k$-th variable in $\mathbf{z}_t$, i.e., $\eta_{kt}(u) \triangleq \log p(z_{t,k} | \mathbf{z}_{t-1}, u)$, and there exists an invertible function $\hat{\mathbf{g}}$ that maps $\mathbf{x}_t$ to $\hat{\mathbf{z}}_t$, i.e., $\hat{\mathbf{z}}_t = \hat{\mathbf{g}}(\mathbf{x}_t)$ such that the components of $\hat{\mathbf{z}}_t$ are mutually independent conditional on $\hat{\mathbf{z}}_{t-1}$. (Sufficient variability) Let*

$$\mathbf{v}_{k,t}(u) \triangleq \left( \frac{\partial^2 \eta_{kt}(u)}{\partial z_{t,k} \partial z_{t-1,1}}, \frac{\partial^2 \eta_{kt}(u)}{\partial z_{t,k} \partial z_{t-1,2}}, ...., \frac{\partial^2 \eta_{kt}(u)}{\partial z_{t,k} \partial z_{t-1,n}} \right)^{\mathsf{T}}, \tag{10}$$

$$\mathring{\mathbf{v}}_{k,t}(u) \triangleq \left( \frac{\partial^3 \eta_{kt}(u)}{\partial z_{t,k}^2 \partial z_{t-1,1}}, \frac{\partial^3 \eta_{kt}(u)}{\partial z_{t,k}^2 \partial z_{t-1,2}}, ...., \frac{\partial^3 \eta_{kt}(u)}{\partial z_{t,k}^2 \partial z_{t-1,n}} \right)^{\mathsf{T}}. \tag{11}$$

$$\mathbf{s}_{kt} \triangleq \left( \mathbf{v}_{kt}(1)^{\mathsf{T}}, ..., \mathbf{v}_{kt}(U)^{\mathsf{T}}, \frac{\partial^2 \eta_{kt}(2)}{\partial z_{t,k}^2} - \frac{\partial^2 \eta_{kt}(1)}{\partial z_{t,k}^2}, ..., \frac{\partial^2 \eta_{kt}(U)}{\partial z_{t,k}^2} - \frac{\partial^2 \eta_{kt}(U-1)}{\partial z_{t,k}^2} \right)^{\mathsf{T}}, \tag{12}$$

$$\mathring{\mathbf{s}}_{kt} \triangleq \left( \mathring{\mathbf{v}}_{kt}(1)^{\mathsf{T}}, ..., \mathring{\mathbf{v}}_{kt}(U)^{\mathsf{T}}, \frac{\partial \eta_{kt}(2)}{\partial z_{t,k}} - \frac{\partial \eta_{kt}(1)}{\partial z_{t,k}}, ..., \frac{\partial \eta_{kt}(U)}{\partial z_{t,k}} - \frac{\partial \eta_{kt}(U-1)}{\partial z_{t,k}} \right)^{\mathsf{T}}. \tag{13}$$

*Suppose $\mathbf{x}_t = \mathbf{g}(\mathbf{z}_t)$ and that the conditional distribution $p(z_{k,t} | \mathbf{z}_{t-1})$ may change across $m$ domains. Suppose that the components of $\mathbf{z}_t$ are mutually independent conditional on $\mathbf{z}_{t-1}$ in each context. Assume that the components of $\hat{\mathbf{z}}_t$ produced by $\hat{\mathbf{g}}$ are also mutually independent conditional on $\hat{\mathbf{z}}_{t-1}$. If the $2n$ function vectors $\mathbf{s}_{k,t}$ and $\mathring{\mathbf{s}}_{k,t}$, with $k = 1, 2, ..., n$, are linearly independent, then $\hat{\mathbf{z}}_t$ is a permuted invertible component-wise transformation of $\mathbf{z}_t$.*

Then, in conjunction with Theorem 1, complete identifiability is achieved for both the domain variables $u_t$ and the independent components $\mathbf{z}_t$. See detailed proof in Appendix S1.3.

**Theorem 2** (Identifiability of the Latent Causal Processes). *Suppose that the observed dataset $\mathcal{D}$ is generated from the nonstationary data generation process as described in Eqs. (1) and (2), which satisfies the conditions in both Theorem 1 and Lemma 1, then the domain variables $u_t$ are identifiable up to label swapping (Def. 2) and latent causal process $\mathbf{z}_t$ are identifiable up to permutation and a component-wise transformation (Def. 3).*

**Discussion on Assumptions** The proof of Theorem 1 relies on several essential assumptions that correspond with human intuition regarding domain transitions. First, the assumption of *separability* posits that if human observers are unable to differentiate between two domains, it is improbable that automated systems will achieve such a distinction. Second, the *variability* assumption requires that the differences in transitions between domains be substantial enough to be perceptible to humans. This often results in changes to the temporal causal structure across domains, indicating that at least one edge in the causal graph must differ between the domains.

The mechanism *sparsity* is a standard assumption that has been previously explored in [33, 19, 18] using sparsity regularization to enforce the sparsity of the estimated function. The assumption of *weakly diverse lossy transitions* is a mild and realistic condition in real-world scenarios, allowing for identical future latent states with differing past states. The *sufficient variability* in Theorem 2 is widely explored and adopted in nonlinear ICA literature [12, 22, 23, 25, 26]. For a more detailed discussion of the feasibility and intuition behind these assumptions, we refer the reader to the Appendix S1.4.

## 4   The CtrlNS Framework

### 4.1   Model Architecture

Our framework builds on VAE [34, 35] architecture, incorporating dedicate modules to handle nonstationarity. It enforces the conditions discussed in Sec. 3 as constraints. As shown in Fig. 2, the framework consists of three primary components: (1) Sparse Transition, (2) Prior Network, and (3) Encoder-Decoder.

**Sparse Transition** The transition module in our framework is designed to estimate transition functions $\{\hat{\mathbf{m}}_u\}_{u=1}^U$ and a clustering function $\hat{\mathcal{C}}$ as specified in Eq. (7). As highlighted in Sec. 3, the primary objective of this module is to model the transitions in the latent space and minimize the empirical transition complexity. To achieve this, we implemented $U$ different transition networks for various $\hat{\mathbf{m}}(\hat{u}_t, \cdot)$ and added sparsity regularization to the transition

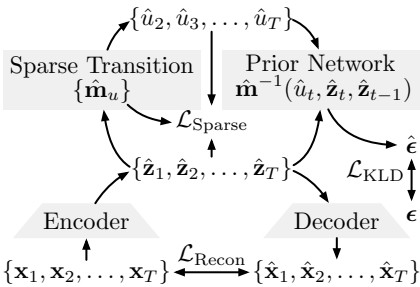

Figure 2: Illustration of **CtrlNS** with (1) Sparse Transition, (2) Prior Network, (3) Encoder-Decoder Module.

functions via a sparsity loss. A gating function with a (hard)-Gumbel-Softmax function was used to generate $\hat{u}_t$, which was then employed to select the corresponding transition network $\hat{\mathbf{m}}_{\hat{u}_t}$. This network was further used to calculate the transition loss, which is explained in detail in Sec. 4.2.

**Prior Network** The Prior Network module aims to effectively estimate the prior distribution $p(\hat{z}_{t,i} \mid \hat{\mathbf{z}}_{t-1}, \hat{u}_t)$. This is achieved by evaluating $p(\hat{z}_t \mid \hat{\mathbf{z}}_{t-1}, \hat{u}_t) = p_{\epsilon_i}\left(\hat{m}_i^{-1}(\hat{u}_t, \hat{z}_{t,i}, \hat{\mathbf{z}}_{t-1})\right)\left|\frac{\partial \hat{m}_i^{-1}}{\partial \hat{z}_{t,i}}\right|$, where $\hat{m}_i^{-1}(\hat{u}_t, \cdot)$ is the learned holistic inverse dynamics model. To ensure the conditional independence of the estimated latent variables, $p(\hat{\mathbf{z}}_t \mid \hat{\mathbf{z}}_{t-1})$, we utilize an isomorphic noise distribution for $\epsilon$ and aggregate all estimated component densities to obtain the joint distribution $p(\hat{\mathbf{z}}_t \mid \hat{\mathbf{z}}_{t-1}, \hat{u}_t)$ as shown in Eq. (14). Given the lower-triangular nature of the Jacobian, its determinant can be computed as the product of its diagonal terms. Detailed derivations is provided in Appendix S3.1.

$$\log p\left(\hat{\mathbf{z}}_t \mid \hat{\mathbf{z}}_{t-1}, \hat{u}_t\right) = \underbrace{\sum_{i=1}^n \log p(\hat{\epsilon}_i \mid \hat{u}_t)}_{\text{Conditional independence}} + \underbrace{\sum_{i=1}^n \log\left|\frac{\partial \hat{m}_i^{-1}}{\partial \hat{z}_{t,i}}\right|}_{\text{Lower-triangular Jacobian}} \tag{14}$$

**Encoder-Decoder** The third component is an Encoder-Decoder module that utilizes reconstruction loss to enforce the invertibility of the learned mixing function $\hat{\mathbf{g}}$. Specifically, the encoder fits the demixing function $\hat{\mathbf{g}}^{-1}$ and the decoder fits the mixing function $\hat{\mathbf{g}}$.

### 4.2 Optimization

The first training objective of **CtrlNS** is to fit the estimated transitions with minimum transition complexity according to Eq. (7):

$$\mathcal{L}_{\text{sparse}} \triangleq \underbrace{\mathbb{E}_{\mathcal{D}} L(\hat{\mathbf{m}}_{\hat{u}_t}(\hat{\mathbf{z}}_{t-1}), \hat{\mathbf{z}}_t)}_{\text{Transition loss}} + \underbrace{\mathbb{E}_{\mathcal{D}} |\widehat{\mathcal{M}}_{\hat{u}}|}_{\text{Sparsity loss}}, \tag{15}$$

where $L(\cdot, \cdot)$ is a regression loss function to fit the transition estimations, and the sparsity loss is approximated via $L_2$ norm of the parameter in the transition estimation functions.

Then the second part is to maximize the Evidence Lower BOund (ELBO) for the VAE framework, which can be written as follows (complete derivation steps are in Appendix S3.2):

$$\text{ELBO} \triangleq \mathbb{E}_{\mathbf{z}_t} \underbrace{\sum_{t=1}^{T} \log p_{\text{data}}(\mathbf{x}_t \mid \mathbf{z}_t)}_{-\mathcal{L}_{\text{Recon}}} + \underbrace{\sum_{t=1}^{T} \log p_{\text{data}}(\mathbf{z}_t \mid \mathbf{z}_{t-1}, u_t) - \sum_{t=1}^{T} \log q_\phi(\mathbf{z}_t \mid \mathbf{x}_t)}_{-\mathcal{L}_{\text{KLD}}} \tag{16}$$

We use mean-squared error for the reconstruction likelihood loss $\mathcal{L}_{\text{Recon}}$. The KL divergence $\mathcal{L}_{\text{KLD}}$ is estimated via a sampling approach since with a learned nonparametric transition prior, the distribution does not have an explicit form. Specifically, we obtain the log-likelihood of the posterior, evaluate the prior $\log p(\hat{\mathbf{z}}_t \mid \hat{\mathbf{z}}_{t-1}, \hat{u}_t)$ in Eq. (14), and compute their mean difference in the dataset as the KL loss: $\mathcal{L}_{\text{KLD}} = \mathbb{E}_{\hat{\mathbf{z}}_t \sim q(\hat{\mathbf{z}}_t \mid \mathbf{x}_t)} \log q(\hat{\mathbf{z}}_t \mid \mathbf{x}_t) - \log p(\hat{\mathbf{z}}_t \mid \hat{\mathbf{z}}_{t-1}, \hat{u}_t)$.

## 5 Experiments

We assessed the identifiability performance of **CtrlNS** on both synthetic and real-world datasets. For synthetic datasets, where we control the data generation process completely, we conducted a comprehensive evaluation. This evaluation covers the full spectrum of unknown nonstationary causal temporal representation learning, including metrics for both domain variables and the latent causal processes. In real-world scenarios, **CtrlNS** was employed in video action segmentation tasks. The evaluation metrics focus on the accuracy of action estimation for each video frame, which reflects the identifiability of domain variables.

### 5.1 Synthetic Experiments on Causal Representation Learning

**Evaluation Metrics** For domain variables, we assessed the *clustering accuracy* (**Acc**) to estimate discrete domain variables $u_t$. As the label order in clustering algorithms is not predetermined, we selected the order that yielded the highest accuracy score. For the latent causal processes, we computed the *mean correlation coefficient* (**MCC**) between the estimated latent variables $\hat{\mathbf{z}}_t$ and the ground truth $\mathbf{z}_t$. The MCC, a standard measure in the ICA literature for continuous variables, assesses the identifiability of the learned latent causal processes. We adjusted the reported MCC values in Table 1 by multiplying them by 100 to enhance the significance of the comparisons.

**Baselines** We compared our method with identifiable nonlinear ICA methods: (1) BetaVAE [36], which ignores both history and nonstationarity information. (2) i-VAE [13] and TCL [10], which leverage nonstationarity to establish identifiability but assume independent factors. (3) SlowVAE [21] and PCL [11], which exploit temporal constraints but assume independent sources and stationary processes. (4) TDRL [23], which assumes nonstationary causal processes but with observed domain indices. (5) HMNLICA [15], which considers the unobserved nonstationary part in the data generation process but does not allow any causally related time-delayed relations. (6) NCTRL [25], which extends HMNLICA to an autoregressive setting to allow causally related time-delayed relations in the latent space but still assumes a Markov chain on the domain variables.

**Result and Analysis** We generate synthetic datasets that satisfy our identifiability conditions in Theorems 1 and 2, detailed procedures are in Appendix S2.1. The primary findings are presented in Table 1. Note: the MCC metric is consistently available in all methods; however, the Acc metric for $u_t$ is only applicable to methods capable of estimating domain variables $u_t$.

Table 1: Experiment results of synthetic dataset on baseline models and the proposed **CtrlNS**. All experiments were conducted using three different random seeds to calculate the average and standard deviation. The best results are highlighted in **bold**.

| $u_t$ | Method | $\mathbf{z}_t$ **MCC** | $u_t$ **Acc** (%) |
|---|---|---|---|
| Ground Truth | TDRL(GT) | $96.93 \pm 0.16$ | - |
| N/A | TCL | $24.19 \pm 0.85$ | |
| | PCL | $38.46 \pm 6.85$ | |
| | BetaVAE | $42.37 \pm 1.47$ | |
| | SlowVAE | $41.82 \pm 2.55$ | - |
| | i-VAE | $81.60 \pm 2.51$ | |
| | TDRL | $53.45 \pm 1.31$ | |
| Estimated | HMNLICA | $17.82 \pm 30.87$ | $13.67 \pm 23.67$ |
| | NCTRL | $47.27 \pm 2.15$ | $34.94 \pm 4.20$ |
| | **CtrlNS** | $\mathbf{96.74 \pm 0.17}$ | $\mathbf{98.21 \pm 0.05}$ |

In the first row of Table 1, we evaluated a recent nonlinear temporal ICA method, TDRL, providing ground truth $u_t$ to establish an upper performance limit for the proposed framework. The high MCC ($> 0.95$) indicates the model's identifiability. Subsequently, the table lists six baseline methods that neglect the nonstationary domain variables, with none achieving a high MCC. The remaining approaches, including our proposed **CtrlNS**, are able to estimate the domain variables $u_t$ and recover the latent variables. In particular, HMNLICA exhibits instability during training, leading to considerable performance variability. This instability stems from HMNLICA's inability to allow time-delayed causal relationships among hidden variables $\mathbf{z}_t$, leading to model training failure when the actual domain variables deviate from the Markov assumption. In contrast, NCTRL, which extends TDRL under the same assumption, demonstrates enhanced stability and performance over HMNLICA by accommodating transitions in $\mathbf{z}_t$. However, since they use incorrect assumption on the nonstationary domain variables, the performance of those methods can be even worse than methods which do not include the domain information. Nevertheless, considering the significant nonstationarity and deviation from the Markov properties, those methods struggled to robustly estimate either the domain variables or the latent causal processes. Compared to all baselines, our proposed **CtrlNS** reliably recovers both $u_t$ (MCC $> 0.95$) and $\mathbf{z}_t$ (Acc $> 95\%$), and the MCC is on par with the upper performance bound when domain variables are given, justifying it effectivess.

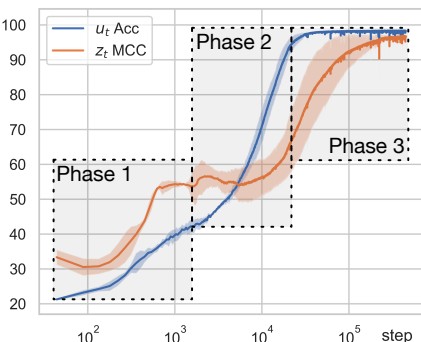

Figure 3: Visualization of three phase training process of **CtrlNS**.

**Detailed Training Analysis** To further validate our theoretical analysis, we present a visualization of the entire training process for **CtrlNS** in Figure 3. It consists of three phases: (1) In Phase 1, the initial estimations for both $u_t$ and $\mathbf{z}_t$ are imprecise. (2) During Phase 2, the accuracy of the estimation of $u_t$ continues to improve, although the quality of the estimation of $\mathbf{z}_t$ remains relatively unchanged compared with Phase 1. (3) In Phase 3, as $u_t$ becomes clearly identifiable, the MCC of $\mathbf{z}_t$ progressively improves, ultimately achieving full identifiability. This three-phase process aligns perfectly with our theoretical predictions. According to Theorem 1, phases 1 and 2 should exhibit suboptimal $\mathbf{z}_t$ estimations, while sparsity constraints can still guide training and improve the accuracy for domain variables $u_t$. Once the accuracy of $u_t$ approaches high, Theorem 2 drives the improvement in MCC for $\mathbf{z}_t$ estimations, leading to the final achievement of full identifiability of both latent causal processes for $\mathbf{z}_t$ and domain variables $u_t$.

## 5.2 Real-world Application on Weakly Supervised Action Segmentation

**Experiment Setup** Our method was tested on the video action segmentation task to estimate actions (domain variables $u_t$). Following [37, 28], we use the same weakly supervised setting utilizing meta-information, such as action order. The evaluation included several metrics: *Mean-over-Frames* (**MoF**), the percentage of correctly predicted labels per frame; *Intersection-over-Union* (**IoU**), defined

Table 2: Real-world experiment result on action segmentation task. We use the reported value for the baseline methods from [28]. Best results are highlighted in **bold**.

| Dataset | Method | MoF | IoU | IoD |
|---|---|---|---|---|
| Hollywood | HMM+RNN [41] | - | 11.9 | - |
| | CDFL [42] | 45.0 | 19.5 | 25.8 |
| | TASL [40] | 42.1 | 23.3 | 33 |
| | MuCon [37] | - | 13.9 | - |
| | ATBA [28] | 47.7 | 28.5 | 44.9 |
| | **CtrlNS (Ours)** | $\mathbf{52.9}_{\pm 3.1}$ | $\mathbf{32.7}_{\pm 1.3}$ | $\mathbf{52.4}_{\pm 1.8}$ |
| CrossTask | NN-Viterbi [43] | 26.5 | 10.7 | 24.0 |
| | CDFL [42] | 31.9 | 11.5 | 23.8 |
| | TASL [40] | 40.7 | 14.5 | **25.1** |
| | POC [44] | 42.8 | 15.6 | - |
| | ATBA [28] | 50.6 | **15.7** | 24.6 |
| | **CtrlNS (Ours)** | $\mathbf{54.0}_{\pm 0.9}$ | $\mathbf{15.7}_{\pm 0.5}$ | $23.6_{\pm 0.8}$ |

as $|I \cap I^*|/|I \cup I^*|$; and *Intersection-over-Detection* (**IoD**), $|I \cap I^*|/|I|$, $I^*$ and $I$ are the ground-truth segment and the predicted segment with the same class.

**Datasets**  Our evaluation used two datasets: Hollywood Extended [38], which includes 937 videos with 16 daily action categories, and CrossTask [39], focusing on 14 of 18 primary tasks related to cooking [40], comprising 2552 videos across 80 action categories.

**Model Design**  Our model is build on top of ATBA [28] method which uses multi-layer transformers as backbone networks. We add our sparse transition module with the sparsity loss function detailed in Sec. 4.2. Specifically, we integrated a temporally latent transition layer into ATBA's backbone, using a transformer layer across time axis for the Hollywood dataset and an LSTM for the CrossTask dataset. To encourage sparsity in the latent transitions, $L_2$ regularization is applied to the weights of the temporally latent transition layer.

**Result and Analysis**  The primary outcomes for real-world applications in action segmentation are summarized in Table 2. Traditional methods based on hidden Markov models, such as HMM+RNN [41] and NN-Viterbi [43], face challenges in these real-world scenarios. This observation corroborates our previous discussions on the limitations of earlier identifiability methods [15, 24, 25], which depend on the Markov assumption for domain variables. Our approach significantly outperforms the baselines in both the Hollywood and CrossTask datasets across most metrics. Especially in the Hollywood dataset, our method outperforms the base ATBA model by quite a large margin. Notably, the Mean-over-Frames (**MoF**) metric aligns well with our identifiability results for domain variables $u_t$. Our method demonstrates substantial superiority in this metric. For Intersection-over-Union (**IoU**) and Intersection-over-Detection (**IoD**), our results are comparable to those of the baseline methods in the CrossTask dataset and show its superiority in the Hollywood dataset. Furthermore, our proposed sparse transition module which aligns with human intuition and is easily integrated into existing methods like a plug-in module, thus further enhancing its impact in real-world scenarios.

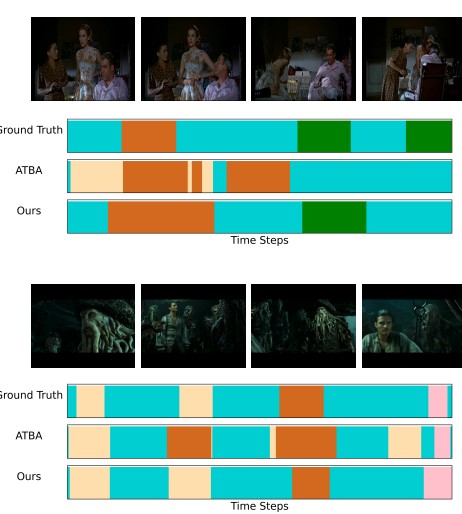

Figure 4: Two illustrative visualizations of the action segmentation task on the Hollywood dataset are presented. The colors represent the ground truth and the predicted action labels for each frame, as produced by the baseline ATBA and our proposed **CtrlNS**.

To make the illustration more straightforward, some example segmentation results from the Hollywood dataset are visualized in Figure 4. By comparing the number of distinct actions and the

segmentation boundaries between our method and the baseline, it is evident that our **CtrlNS** estimates the actions more accurately, demonstrating improved performance.

Table 3: Ablation study on sparse transition module in Hollywood dataset.

| Method | MoF | IoU | IoD |
|---|---|---|---|
| **CtrlNS** | **52.9** | **32.7** | **52.4** |
| - Complexity | 50.5 | 31.5 | 51.5 |
| - Module | 47.7 | 28.5 | 44.9 |

**Abalation Study** Furthermore, we conducted an ablation study on the sparse transition module, as detailed in Table 3. In this study, we test on a subset of Hollywood dataset for computational efficiency. For methods we compared, "- Complexity" refers to the configuration where we retain the latent transition layers but omit the sparse transition complexity regularization term from these layers, and "- Module" indicates the removal of the entire sparse transition module, effectively reverting the model to the baseline ATBA model. The comparative results in Table 3 demonstrate that both the dedicated design of the sparse transition module and the complexity regularization term enhance the performance.

# 6 Conclusion

In this study, we developed a comprehensive identifiability theory tailored for general sequential data influenced by nonstationary causal processes under unspecified distributional changes. We then introduced **CtrlNS**, a principled approach to recover both latent causal variables with their time-delayed causal relations, as well as determining the values of domain variables from observational data without relying on distributional or structural prior knowledge. Our experimental results demonstrate that **CtrlNS** can reliably estimate the domain indices and recover the latent causal process. And such module can be easily adapted to handle real-world scenarios such as action segmentation task.

# 7 Limitations

As noted in Sec. 3.2, our main theorem relies on the condition that causal graphs among different domains must be distinct. Although our experiments indicate that this assumption is generally sufficient, there are scenarios in which it may not hold, meaning that the transition causal graphs are identical for two different domains, but the actual transition functions are different. We have addressed this partially through an extension to the mechanism variability assumption to higher-order cases (Corollary 1). However, dealing with situations where transition graphs remain the same across all higher orders remains a challenge. We acknowledge this as a limitation and suggest it as an area for future exploration. We also observed that the random initialization of the nonlinear ICA framework can influence the total number of epochs needed to achieve identifiability, as illustrated in Figure 3. Also, for the computational efficiency, the TDRL framework we adopted involves a prior network that calculated each dimension in the latent space one by one, thus making the training efficiency suboptimal. Since this is not directly related to major claim which is our sparse transition design, we acknowledge this as a limitation and leave it for future work.

# 8 Boarder Impacts

This work proposes a theoretical analysis and technical methods to learn the causal representation from time-series data, which facilitate the construction of more transparent and interpretable models to understand the causal effect in the real world. This could be beneficial in a variety of sectors, including healthcare, finance, and technology. In contrast, misinterpretations of causal relationships could also have significant negative implications in these fields, which must be carefully done to avoid unfair or biased predictions.

# 9 Acknowledgment

The authors would like to thank the anonymous reviewers for helpful comments and suggestions during the reviewing process. The authors would also like to acknowledge the support from NSF Award No. 2229881, AI Institute for Societal Decision Making (AI-SDM), the National Institutes of Health (NIH) under Contract R01HL159805, and grants from Salesforce, Apple Inc., Quris AI, and Florin Court Capital.

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

*Supplement to*

# "Causal Temporal Representation Learning with Nonstationary Sparse Transition"

Appendix organization:

# S1 Identifiability Theory

## S1.1 Proof for Theorem 1

We divide the complete proof into two principal steps:

1. Firstly, assuming access to the optimal mixing function estimation $\hat{\mathbf{g}}^*$, we demonstrate that under the conditions in our theorem, the estimated clustering result will align with the ground truth up to label swapping. This alignment is due to the transition complexity with optimal $\hat{u}_t^*$ and $\hat{\mathbf{g}}^*$ being strictly lower than that with non-optimal $\hat{u}_t$ but still optimal $\hat{\mathbf{g}}^*$.

2. Secondly, we generalize the results of the first step to cases where the mixing function estimation $\hat{\mathbf{g}}$ is suboptimal. We establish that for any given clustering assignment, whether optimal or not, a suboptimal mixing function estimation $\hat{\mathbf{g}}$ can not result in a lower transition complexity. Thus, the transition complexity in scenarios with non-optimal $\hat{\mathbf{g}}$ will always be at least as high as in the optimal case.

From those two steps, we conclude that the global minimum transition complexity can only be attained when the estimation of domain variables $\hat{u}_t$ is optimal, hence ensuring that the estimated clustering must match the ground truth up to label swapping. It is important to note that this condition alone does not guarantee the identifiability of the mixing function $\mathbf{g}$. Because a setting with optimal $\hat{u}_t^*$ and a non-optimal $\hat{\mathbf{g}}$ may exhibit equivalent transition complexity to the optimal scenario, but it does not compromise our proof for the identifiability of domain variables $u_t$. Further exploration of the mixing function's identifiability $\mathbf{g}$ is discussed in Theorem 2 in the subsequent section.

### S1.1.1 Identifiability of $\mathcal{C}$ under optimal $\hat{\mathbf{g}}^*$

We fist introduce a lemma for this case when we can access an optimal mixing function estimation $\hat{\mathbf{g}}^*$.

**Lemma S1** (Identifiability of $\mathcal{C}$ under optimal $\hat{\mathbf{g}}^*$). *In addition to the assumptions in Theorem 1, assume that we can also access an optimal estimation of $\mathbf{g}$, denoted by $\hat{\mathbf{g}}^*$, in which the estimated $\hat{\mathbf{z}}_t$ is an invertible, component-wise transformation of a permuted version of $\mathbf{z}_t$. Then the estimated clustering $\hat{\mathcal{C}}$ must match the ground truth up to label swapping.*

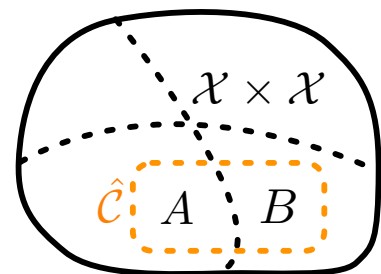

Figure S1: Illustration of $\hat{\mathcal{C}}$ incorrectly assigning two different domain subsets of inputs $A$ and $B$ into the same $\hat{u}$. The black lines represent the ground truth partition of $\mathcal{C}$ and the orange line represent the incorrect domain partition for set $A$ and $B$.

*Proof.* In the first case we deal with optimal estimation $\hat{\mathbf{g}}^*$ in which the estimated $\hat{\mathbf{z}}_t$ is an invertible, component-wise transformation of a permuted version of $\mathbf{z}_t$, but inaccurate estimated version of $\hat{\mathcal{C}}$, Consider the following example (Figure S1):

With slight abuse of notation, we use $\mathcal{C}(A)$ to represent the domain assigned by $\mathcal{C}$ to all elements in $A$, and all elements in $A$ have the same assignment. The same argument applies to $B$.

Then, for an estimated $\hat{\mathcal{C}}$, if it incorrectly assigns two subsets of input $A$ and $B$ to the same $\hat{u}$ (Figure S1 orange circle), i.e.,

$$\mathcal{C}(A) = i \neq j = \mathcal{C}(B) \quad \text{but} \quad \hat{\mathcal{C}}(A) = \hat{\mathcal{C}}(B) = k. \tag{17}$$

Note that if the ground truth $\mathcal{C}$ gives a consistent assignment for $A$ and $B$ but estimated $\hat{\mathcal{C}}$ gives diverse assignments, i.e.

$$\hat{\mathcal{C}}(A) = i \neq j = \hat{\mathcal{C}}(B) \quad \text{but} \quad \mathcal{C}(A) = \mathcal{C}(B) = k, \tag{18}$$

it is nothing but further splitting the ground truth assignment in a more fine-grained manner. This scenario does not break the boundaries of the ground truth assignments. Consider two cases in the estimation process:

1. If the number of allowed regimes or domains exceeds that of the ground truth, such more fine-grained assignment is allowed. The ground truth can then be easily recovered by merging domains that share identical Jacobian supports.

2. If the number of regimes or domains matches the ground truth, it can be shown that the inconsistent scenario outlined in Eq. (17) must occur.

Given that these considerations do not directly affect our approach, they are omitted from further discussion for brevity.

$$\mathcal{M}_i = \begin{bmatrix} \textcolor{red}{0} & 0 & 1 & 0 \\ 1 & 1 & 0 & 0 \\ 0 & 0 & 0 & 1 \\ 0 & 0 & 1 & 0 \end{bmatrix} \quad \mathcal{M}_j = \begin{bmatrix} \textcolor{red}{1} & 0 & 1 & 0 \\ 1 & 1 & 0 & 0 \\ 0 & 0 & 0 & 1 \\ 0 & 0 & 1 & 0 \end{bmatrix} \quad \widehat{\mathcal{M}}_k = \begin{bmatrix} \textcolor{red}{1} & 0 & 1 & 0 \\ 1 & 1 & 0 & 0 \\ 0 & 0 & 0 & 1 \\ 0 & 0 & 1 & 0 \end{bmatrix}$$

Figure S2: Comparison of matrices $\mathcal{M}_i$, $\mathcal{M}_j$, and $\widehat{\mathcal{M}}_k$. The elements in red highlight the differences between them.

Then considering the case in Eq. 17, the estimated transition must cover the functions from both $A$ and $B$, then the learned transition $\hat{\mathbf{m}}_k$ must have Jacobian $\mathbf{J}_{\hat{\mathbf{m}}_k}$ with support matrix $\widehat{\mathcal{M}}_k = \mathcal{M}_i + \mathcal{M}_j$ which is the binary addition of $\mathcal{M}_i$ and $\mathcal{M}_j$, such that for all indices in $\mathcal{M}_i, \mathcal{M}_j$ if any of these two is 1, then the corresponding position in $\widehat{\mathcal{M}}_k$ must be 1. That is because if that is not the case, for example, the $(a, b)$-th location for $\mathcal{M}_i$, $\mathcal{M}_j$, and $\widehat{\mathcal{M}}_k$ are 1, 0, and 0. Then we can easily find an input region for the $(a, b)$-th location such that a small perturbation can lead to changes in $\mathbf{m}_i$ but not in $\mathbf{m}_j$ nor $\hat{\mathbf{m}}_k$, which makes $\hat{\mathbf{m}}_k$ unable to fit all of the transitions in $A \cup B$ which cause contradiction. See the three matrices in Figure S2 for an illustrative example.

By assumption iii, since all those support matrix differ at least one spot, which means the estimated version is not smaller than the ground truth.

$$|\widehat{\mathcal{M}}_k| \geq |\mathcal{M}_j| \quad \text{and} \quad |\widehat{\mathcal{M}}_k| \geq |\mathcal{M}_i|, \tag{19}$$

and the equality cannot hold true at the same time.

Then from Assumption (i), the expected estimated transition complexity can be expressed as:

$$\mathbb{E}_{\mathcal{D}}|\widehat{\mathcal{M}}_{\hat{u}}| = \int_{\mathcal{X} \times \mathcal{X}} p_{\mathcal{D}}(\mathbf{x}_{t-1}, \mathbf{x}_t) \cdot |\widehat{\mathcal{M}}_{\hat{\mathcal{C}}(\mathbf{x}_{t-1}, \mathbf{x}_t)}| \, d\mathbf{x}_{t-1} \, d\mathbf{x}_t. \tag{20}$$

Similarly for ground truth one:

$$\mathbb{E}_{\mathcal{D}}|\mathcal{M}_u| = \int_{\mathcal{X} \times \mathcal{X}} p_{\mathcal{D}}(\mathbf{x}_{t-1}, \mathbf{x}_t) \cdot |\mathcal{M}_{\mathcal{C}(\mathbf{x}_{t-1}, \mathbf{x}_t)}| \, d\mathbf{x}_{t-1} \, d\mathbf{x}_t. \tag{21}$$

Let us focus on the integral of the region $A \cup B$, the subset of $\mathcal{X} \times \mathcal{X}$ mentioned above. If for some area $p_{\mathcal{D}}(\mathbf{x}_{t-1}, \mathbf{x}_t) = 0$, then the clustering under this area is ill defined since there is no support from data. Hence we only need to deal with supported area. For area that $p_{\mathcal{D}}(\mathbf{x}_{t-1}, \mathbf{x}_t) > 0$ and from Eq. (19) the equality cannot hold true at the same time, then the estimated version of the integral is strictly larger than the ground-truth version for any inconsistent clustering as indicated in Eq. (17).

For the rest of regions in $\mathcal{X} \times \mathcal{X}$, any incorrect cluster assignment will further increase the $\mathcal{M}$ with same reason as discussed above, then the estimated complexity is strictly larger than the ground truth complexity:

$$\mathbb{E}_{\mathcal{D}}|\widehat{\mathcal{M}}_{\hat{u}}| > \mathbb{E}_{\mathcal{D}}|\mathcal{M}_u|. \tag{22}$$

But assumption (ii) requires that the estimated complexity be less than or equal to the ground truth. Contradiction! Hence, the estimated $\hat{\mathcal{C}}$ must match the ground truth up to label swapping. $\qquad \square$

### S1.1.2 Identifiability of $\mathcal{C}$ under arbitrary g

Now we can leverage the conclusion in Lemma S1 to show the identifiability of domain variables under arbitrary mixing function estimation.

**Theorem S1** (Identifiability of Domain Variables). *Suppose that the data $\mathcal{D}$ are generated from the nonstationary data generation process as described in Eqs.* (1) *and* (2). *Suppose the transitions are weakly diverse lossy (Def. 6) and the following assumptions hold:*

i. *(Mechanism Separability) There exists a ground truth mapping $\mathcal{C} : \mathcal{X} \times \mathcal{X} \to \mathcal{U}$ determined the real domain indices, i.e., $u_t = \mathcal{C}(\mathbf{x}_{t-1}, \mathbf{x}_t)$.*

ii. *(Mechanism Sparsity) The estimated transition complexity on dataset $\mathcal{D}$ is less than or equal to ground truth transition complexity, i.e., $\mathbb{E}_{\mathcal{D}} |\widehat{\mathcal{M}}_{\hat{u}}| \leq \mathbb{E}_{\mathcal{D}} |\mathcal{M}_u|$.*

iii. *(Mechanism Variability) Mechanisms are sufficiently different. For all $u \neq u'$, $\mathcal{M}_u \neq \mathcal{M}_{u'}$ i.e. there exists index $(i, j)$ such that $[\mathcal{M}_u]_{i,j} \neq [\mathcal{M}_{u'}]_{i,j}$.*

*Then the domain variables $u_t$ is identifiable up to label swapping (Def. 2).*

*Proof.* To demonstrate the complete identifiability of $\mathcal{C}$, independent of the estimation quality of $\mathbf{g}$, we must show that for any arbitrary estimation $\hat{\mathcal{C}} \neq \sigma(\mathcal{C})$, the induced $\widehat{\mathcal{M}}_{\hat{u}}$ for inaccurate estimation $\hat{\mathbf{g}}$ has a transition complexity at least as high as in the optimal $\hat{\mathbf{g}}^*$ case. If this holds, from Lemma S1, we can conclude that the transition complexity of optimal $\hat{\mathcal{C}}^* = \sigma(\mathcal{C})$ and optimal $\hat{\mathbf{g}}^*$ is strictly smaller than any non-optimal $\hat{\mathcal{C}}$ and any $\hat{\mathbf{g}}$.

Suppose the estimated decoder and corresponding latent variables are $\hat{\mathbf{g}}$ and $\hat{\mathbf{z}}_t$, respectively, then the following relation holds:

$$\hat{\mathbf{g}}^*(\mathbf{z}_t) = \hat{\mathbf{g}}(\hat{\mathbf{z}}_t). \tag{23}$$

Since $\hat{\mathbf{g}}$ is invertible, by composing $\hat{\mathbf{g}}^{-1}$ on both sides, we obtain:

$$\hat{\mathbf{g}}^{-1} \circ \hat{\mathbf{g}}^*(\mathbf{z}_t) = \hat{\mathbf{g}}^{-1} \circ \hat{\mathbf{g}}(\hat{\mathbf{z}}_t). \tag{24}$$

Let

$$\mathbf{h} := \hat{\mathbf{g}}^{-1} \circ \hat{\mathbf{g}}^*, \tag{25}$$

we then have:

$$\mathbf{h}(\hat{\mathbf{z}}_t^*) = \hat{\mathbf{z}}_t. \tag{26}$$

We aim to demonstrate that under this transformation, if $\mathbf{h}$ is not a permutation and component-wise transformation, the introduced transition complexity among estimated $\hat{\mathbf{z}}$ will not be smaller than the optimal $\hat{\mathbf{g}}^*$.

**Proposition S1.** *Suppose $|\widehat{\mathcal{M}}| < |\widehat{\mathcal{M}}^*|$, then for any permutation $\sigma$ mapping the indices of the dimensions from $\widehat{\mathcal{M}}$ to $\widehat{\mathcal{M}}^*$, there must exist an index pair $(i, j)$ such that $\widehat{\mathcal{M}}_{i,j} = 0$ and $\widehat{\mathcal{M}}^*_{\sigma(i),\sigma(j)} = 1$.*

*Proof.* An intuitive explanation for this proposition involves the construction of a directed graph $G_{\widehat{\mathcal{M}}^*} = (V_{\widehat{\mathcal{M}}^*}, E_{\widehat{\mathcal{M}}^*})$, where $V_{\widehat{\mathcal{M}}^*} = \{1, 2, \ldots, n\}$ and $E_{\widehat{\mathcal{M}}^*} = \{(i, j) \mid \widehat{\mathcal{M}}^*_{i,j} = 1\}$. A similar construction can be made for $G_{\widehat{\mathcal{M}}}$. It is straightforward that $|\widehat{\mathcal{M}}^*| = |E_{\widehat{\mathcal{M}}^*}|$, which represents the number of edges. Consequently, $|\widehat{\mathcal{M}}| < |\widehat{\mathcal{M}}^*|$ implies that $G_{\widehat{\mathcal{M}}^*}$ has more edges than $G_{\widehat{\mathcal{M}}}$. Since there is no pre-defined ordering information for the nodes in these two graphs, if we wish to compare their edges, we need to first establish a mapping. However, if $|E_{\widehat{\mathcal{M}}}| < |E_{\widehat{\mathcal{M}}^*}|$, no matter how the mapping $\sigma$ is constructed, there must be an index pair $(i, j)$ such that $(i, j) \notin E_{\widehat{\mathcal{M}}}$ but $(\sigma(i), \sigma(j)) \in E_{\widehat{\mathcal{M}}^*}$. Otherwise, if such an index pair does not exist, the total number of edges in $G_{\widehat{\mathcal{M}}}$ would necessarily be greater than or equal to that in $G_{\widehat{\mathcal{M}}^*}$, contradicting the premise that $|\widehat{\mathcal{M}}| < |\widehat{\mathcal{M}}^*|$. $\square$

**Lemma S2** (Non-decreasing Complexity). *Suppose transitions are weakly diverse lossy as defined in Def. 6 and an invertible transformation $\mathbf{h}$ maps the optimal estimation $\hat{\mathbf{z}}_t^*$ to the estimated $\hat{\mathbf{z}}_t$, and it is neither a permutation nor a component-wise transformation. Then, the transition complexity on the estimated $\hat{\mathbf{z}}_t$ is not lower than that on the optimal $\hat{\mathbf{z}}_t^*$, i.e.,*

$$|\widehat{\mathcal{M}}| \geq |\widehat{\mathcal{M}}^*|.$$

*Proof.* The entire proof is based on contradiction. In Figure S3, we provide an illustrative example. Note that the mapping from ground truth $\mathbf{z}_t$ to optimal estimation $\hat{\mathbf{z}}_t^*$ is a permutation and element-wise transformation, it does not include mixing, and hence $e_i$ exists if and only if $\hat{e}_i^*$ exists. Therefore, $|\widehat{\mathcal{M}}^*| = |\mathcal{M}|$. The core of the proof requires us to demonstrate that $|\widehat{\mathcal{M}}| < |\mathcal{M}|$ cannot be true.

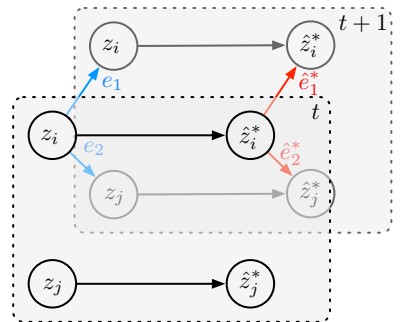
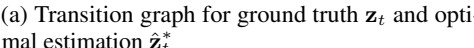
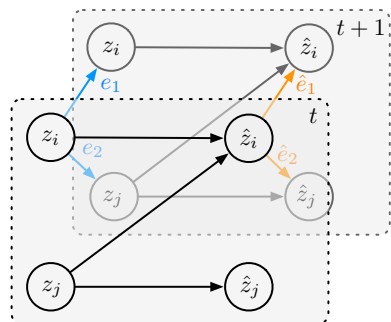

(a) Transition graph for ground truth $\mathbf{z}_t$ and optimal estimation $\hat{\mathbf{z}}_t^*$

(b) Transition graph for ground truth $\mathbf{z}_t$ and arbitrary estimation $\hat{\mathbf{z}}_t$

Figure S3: A partial observation of the transition graph among ground truth $\mathbf{z}_t$, optimal estimation $\hat{\mathbf{z}}_t^*$ and arbitrary estimation $\hat{\mathbf{z}}_t$. For brevity, the index permutation is assumed to be identity, i.e., $\sigma(i) = i$.

Suppose the transitions are weakly diverse lossy as defined in Def. 6, then for each edge $z_{t,i} \rightarrow z_{t+1,j}$ in the transition graph, there must be a region of $z_{t,i}$ such that only $z_{t+1,j}$ is influenced by $z_{t,i}$. Consequently, the corresponding $\hat{z}_{t+1,j}$ and $\hat{z}_{t+1,i}$ are not independent, since no mixing process can cancel the influence of $z_{t,i}$. Therefore, the edge $\hat{z}_{t,i} \rightarrow \hat{z}_{t+1,j}$ in the estimated graph must exist.

Note that without the weakly diverse lossy transition assumption, this argument may not hold. For example, if $\hat{z}_{t+1,j}$ can be expressed as a function that does not depend on $z_{t,i}$, then even though the edge $z_{t,i} \rightarrow z_{t+1,j}$ exists, the estimated edge $\hat{z}_{t,i} \rightarrow \hat{z}_{t+1,j}$ may not exist. This could occur if, after the transformation $\mathbf{h}$, the influences in different paths from $z_{t,i}$ to $\hat{z}_{t+1,j}$ cancel out with each other.

**Necessity Example** An example that violates the assumption is as follows:

$$z_{t+1,i} = z_{t,i} + \epsilon_{t+1,i}$$
$$z_{t+1,j} = z_{t,i} + \epsilon_{t+1,j}$$
$$\hat{z}_i = z_i$$
$$\hat{z}_j = z_i - z_j$$

Here, the mapping from $\mathbf{z}$ to $\hat{\mathbf{z}}$ is invertible. Writing down the mapping from $\hat{\mathbf{z}}_t$ to $\hat{\mathbf{z}}_{t+1}$, particularly for $\hat{z}_{t+1,j}$, yields:

$$\hat{z}_{t+1,j} = (z_{t,i} + \epsilon_{t+1,i}) - (z_{t,i} + \epsilon_{t+1,j})$$
$$= \epsilon_{t+1,i} - \epsilon_{t+1,j}$$

Clearly, this is independent of $\hat{z}_{t,i}$. Hence, in this scenario, the edge on the estimated graph does not exist. This explains the necessity for the weakly diverse lossy transition assumption. Furthermore, it can be seen that violating the weakly diverse lossy transition assumption would require a very specific design, such as the transition in an additive noise case and the transition on $z$ being linear, which is usually not the case in real-world scenarios. Generally, this requires that the influences from different paths from $z_{t,i}$ to $\hat{z}_{t+1,j}$ cancel each other out, a condition that is very challenging to fulfill in practical settings.

**Permutation Indexing** One may also ask about the permutation of the index between $\mathbf{z}_t$ and $\hat{\mathbf{z}}_t$. Since the transformation $\mathbf{h}$ is invertible, the determinant of the Jacobian should be nonzero, implying the existence of a permutation $\sigma$ such that

$$(i, \sigma(i)) \in \text{supp}(\mathbf{J_h}), \ \forall i \in [n].$$

Otherwise, if there exists an $i$ such that $[\mathbf{J_h}]_{i,\cdot} = \mathbf{0}$ or $[\mathbf{J_h}]_{\cdot,i} = \mathbf{0}$, such a transformation cannot be invertible. We can utilize this permutation $\sigma$ to pair the dimensions in $\mathbf{z}_t$ and $\hat{\mathbf{z}}_t$.

Since each ground-truth edge is preserved in the estimated graph, by Proposition S1, the inequality $|\widehat{\mathcal{M}}| < |\widehat{\mathcal{M}}^*|$ cannot hold true. Thus, the lemma is proved. $\qquad \square$

Then, according to this lemma, the transition complexity $|\widehat{\mathcal{M}}_{\hat{u}}|$ of the learned $\hat{\mathbf{m}}_{\hat{u}}$ should be greater than or equal to $|\widehat{\mathcal{M}}_{\hat{u}}^*|$, which is the complexity when using an accurate estimation of $\hat{\mathbf{g}}^*$. This relationship can be expressed as follows:

$$|\widehat{\mathcal{M}}_{\hat{u}}| \geq |\widehat{\mathcal{M}}_{\hat{u}}^*|.$$

By Lemma S1, the expected complexity of the estimated model $\mathbb{E}_{\mathcal{D}}|\widehat{\mathcal{M}}_{\hat{u}}^*|$ is strictly larger than that of the ground truth $\mathbb{E}_{\mathcal{D}}|\mathcal{M}_u|$. This implies the following inequality:

$$\mathbb{E}_{\mathcal{D}}|\widehat{\mathcal{M}}_{\hat{u}}| \geq \mathbb{E}_{\mathcal{D}}|\widehat{\mathcal{M}}_{\hat{u}}^*| > \mathbb{E}_{\mathcal{D}}|\mathcal{M}_u|. \tag{27}$$

However, Assumption (ii) requires that the estimated complexity must be less than or equal to the ground-truth complexity, leading to a contradiction. This contradiction implies that the estimated $\hat{\mathcal{C}}$ must match the ground truth up to label swapping. Consequently, this supports the conclusion of Theorem 1. □

## S1.2   Proof of Corollary 1

**Corollary S1** (Identifiability under Function Variability). *Suppose the data $\mathcal{D}$ is generated from the nonstationary data generation process described in* (1) *and* (2)*. Assume the transitions are weakly diverse lossy (Def. 6), and the mechanism separability assumption i along with the following assumptions hold:*

  v. *(Mechanism Function Variability) Mechanism Functions are sufficiently different. There exists $K \in \mathbb{N}$ such that for all $u \neq u'$, there exists $k \leq K$, $\mathcal{M}_u^k \neq \mathcal{M}_{u'}^k$ i.e. there exists index $(i,j)$ such that $\left[\mathcal{M}_u^k\right]_{i,j} \neq \left[\mathcal{M}_{u'}^k\right]_{i,j}$.*

  vi. *(Higher Order Mechanism Sparsity) The estimated transition complexity on dataset $\mathcal{D}$ is no more than ground truth transition complexity,*

$$\mathbb{E}_{\mathcal{D}} \sum_{k=1}^K |\widehat{\mathcal{M}}_{\hat{u}}^k| \leq \mathbb{E}_{\mathcal{D}} \sum_{k=1}^K |\mathcal{M}_u^k|. \tag{28}$$

*Then the domain variables $u_t$ are identifiable up to label swapping (Def. 2).*

*Proof.* With a strategy similar to the proof of Theorem 1, we aim to demonstrate that using an incorrect cluster assignment $\hat{\mathcal{C}}$ will result in $\sum_{k=1}^K |\mathcal{M}_{\hat{u}}^k|$ being higher than the ground truth, thereby still enforcing the correct $u_t$.

Differing slightly from the approach in Theorem 1, in this setting, we will first demonstrate that under any arbitrary $\hat{\mathcal{C}}$ assignment, the estimated complexity is no lower than the complexity in the ground truth, i.e., $\sum_{k=1}^K |\widehat{\mathcal{M}}^k| \geq \sum_{k=1}^K |\mathcal{M}^k|$.

First, we address the scenario where two different domains have the same transition graph but with different functions, as otherwise, the previous lemma S1 still applies. In cases where the same transition causal graph exists but the functions differ, assumption v indicates that there exists an integer $k$ such that $\mathcal{M}_u^k \neq \mathcal{M}_{u'}^k$, meaning the ground truth support matrices are different. However, due to incorrect clustering, the learned transition must cover both cases. To substantiate this claim, we need to first introduce an extension of the non-decreasing complexity lemma.

**Lemma S3** (Non-decreasing Complexity under Mechanism Function Variability). *Suppose there exists an invertible transformation $\mathbf{h}$ which maps the ground truth $\mathbf{z}_t$ to the estimated $\hat{\mathbf{z}}_t$, and it is neither a permutation nor a component-wise transformation. Then, the transition complexity on the estimated $\hat{\mathbf{z}}_t$ is not lower than that on the ground truth $\mathbf{z}_t$, i.e.,*

$$\sum_{k=1}^K |\widehat{\mathcal{M}}^k| \geq \sum_{k=1}^K |\mathcal{M}^k|.$$

*Proof.* We can extend the notation of the edges $e$ to the higher-order case $e^k$ to represent the existence of a non-zero value for the $k$-th order partial derivative $\frac{\partial^k m_i}{\partial z_j^k}$. Under the weakly diverse lossy transition assumption, it is always possible to find a region where the influence in $e^k$ cannot be canceled in $\hat{e}^k$. In this region, the mapping from $z_i$ to $\hat{z}_i$ can be treated as a component-wise transformation, since the influence of $\mathbf{z}$ other than $z_i$ is zero due to the lossy transition assumption. It is important to note that there is also an indexing permutation issue between $z_i$ and $\hat{z}_i$; the same argument in the permutation indexing part of the proof of lemma S3 applies.

Since $\mathcal{M}^k$ represents the support of the $k$-th order partial derivative, this implies that $[\mathcal{M}^k]_{i,j} = 1$ implies $[\mathcal{M}^{k'}]_{i,j} = 1$ for all $k' \leq k$. We aim to show that if for the transition behind edge $z_{t,j} \to z_{t+1,i}$, there exists a $K$ such that $[\mathcal{M}_u^k]_{i,j}$ are different for two domains, then one of them must be a polynomial with order $K - 1$. For this domain, $[\mathcal{M}_u^k]_{i,j} = 1$ when $k = 1, 2, \ldots, K - 1$ and $[\mathcal{M}_u^k]_{i,j} = 0$ when $k \geq K$.

To demonstrate that the non-decreasing complexity holds, we need to show that after an invertible transformation $h$ to obtain the estimated version, $[\widehat{\mathcal{M}}_u^k]_{i,j}$ cannot be zero for $k < K - 1$, which can be shown with the following proposition.

**Proposition S2.** *Suppose $f$ is a polynomial of order $k$ with respect to $x$. Then, for any invertible smooth function $h$, the transformed function $\hat{f} := h^{-1} \circ f \circ h$ cannot be expressed by a polynomial of order $k'$, when $k' < k$.*

*Proof.* Let's prove it by contradiction. Suppose $\hat{f} := h^{-1} \circ f \circ h$ can be expressed as a polynomial of order $k' < k$. It follows that the function $\hat{f}(x) = C$ has $k'$ roots (repeated roots are allowed), since $h$ is invertible. Therefore, $h \circ \hat{f}(x) = h(C)$ also has the same number of $k'$ roots. By definition, $h \circ \hat{f} = f \circ h$, which means $f \circ h = h(C)$ has $k'$ roots. However, since $h$ is invertible, or equivalently it is monotonic, the equation $f \circ h = h(C)$ having $k'$ roots implies that $f(x) = C'$ has roots $k'$. Yet, since $f$ is a polynomial of order $k$, it must have $k$ roots, contradicting the fundamental theorem of algebra, which means that they cannot have the same number of roots. Hence, the proposition holds. $\square$

The advantage of support matrix analysis is that, provided there exists at least one region where the support matrix is non-zero, the global version on the entire space will also be non-zero. Based on the definition of diverse lossy transition in Def. 6, it is always possible to identify such a region where for an edge $z_{t,i} \to z_{t+1,j}$, the mapping from $z_{t+1,j}$ to $z_{t+1,j}$ can be treated as a component-wise relationship. This is because no other variables besides $z_{t+1,j}$ change in conjunction with $z_{t,i}$ to cancel the effect. Therefore, proposition S2 applies, and as a result, the complexity is nondecreasing. Thus, the lemma is proved. $\square$

With this lemma, we have shown that for an arbitrary incorrect domain partition result, the induced ground-truth transition complexity is preserved after the invertible transformation $\mathbf{h}$. This partition effectively combines two regions, as illustrated in Figures S1 and S4. Consequently, the transition complexity has the following relationship:

$$\widehat{\mathcal{M}}_{\hat{u}}^k = \mathcal{M}_u^k + \mathcal{M}_{u'}^k.$$

Here, $u$ and $u'$ represent the ground-truth values of the domain variables, and $\hat{u}$ denotes the estimated version, defined as the binary addition of the two ground truths.

By assumption v, the two ground truth transitions' complexity $\mathcal{M}_u^k, \mathcal{M}_{u'}^k$ are different, then with the same arguments in the proof of the lemma S1, we can show that the expected transition complexity with wrong domain assignment over the whole dataset is strictly larger than the ground truth complexity with correct domain assignment. And it is easy to see that when the estimated latent variables are equal to the ground truth, $\hat{\mathbf{z}}_t = \mathbf{z}_t$ then the lower bound is achieved when the estimated domains are accurate. Note that this argument is not a sufficient condition to say that the estimated $\hat{\mathbf{z}}_t$ is exactly the ground truth $\mathbf{z}_t$ or an optimal estimation of it, since there can be other formats of mapping from $\mathbf{z}_t$ to $\hat{\mathbf{z}}_t$ that generate the same complexity. But this is sufficient to prove that by pushing the complexity to small, the domain variables $u_t$ must be recovered up to label swapping. This concludes the proof. $\square$

## S1.3 Proof of Theorem 2

**Theorem S2** (Identifiability of the Latent Causal Processes). *Suppose that the data $\mathcal{D}$ is generated from the nonstationary data generation process* (1)*,* (2)*, which satisfies the conditions in Theorem 1 and Lemma 1, then the domain variables $u_t$ are identifiable up to label swapping (Def. 2) and latent variables $\mathbf{z}_t$ are identifiable up to permutation and a component-wise transformation (Def. 3).*

*Proof.* From Theorem 1, the domain variables $u_t$ are identifiable up to label swapping, and then use the estimated domain variables in Lemma 1, the latent causal processes are also identifiable, that is, $\mathbf{z}_t$ are identifiable up to permutation and a component-wise transformation. $\qquad\square$

## S1.4 Discussion on Assumptions

### S1.4.1 Mechanism Separability

Note that we assume that there exists a ground truth mapping $\mathcal{C} : \mathcal{X} \times \mathcal{X} \to \mathcal{U}$, gives a domain index based on $\mathbf{x}_{t-1}, \mathbf{x}_t$. The existence of such mapping means that the human can tell what the domain is based on two consecutive observations. If two observations are not sufficient, then it can be modified to have more observation steps as input, for example $\mathbf{x}_{\leq t}$ or even full sequence $\mathbf{x}_{[1:T]}$. If the input has future observation, which means that $\mathbf{x}_{>t}$ is included, then this is only valid for sequence understanding tasks in which the entire sequence will be visible to the model when analyzing the time step $t$. For prediction tasks or generation tasks, further assumptions on $\mathcal{C}$ such as the input only contains $\mathbf{x}_{<t}$ should be made, which will be another story. Those variants are based on specific application scenarios and not directly affect our theory, for brevity, let us assume the two-step case.

### S1.4.2 Mechanism Sparsity

This is a rather intuitive assumption in which we introduce some form of sparsity in the transitions, and our task is to ensure that the estimated transition maintains this sparsity pattern. This requirement is enforced by asserting an equal or lower transition complexity as defined in Assumption ii. Similar approaches, grounded in the same intuition, are also explored in the reinforcement learning setting, as discussed in works by Lachapelle et al. [18] and Hwang et al. [45]. The former emphasizes the identifiability result of the independent components, which necessitates additional assumptions. In contrast, the latter focuses on the RL scenario, requiring the direct observation of the latent variables involved in the dynamics, which leaves significant challenges in real-world sequence-understanding tasks, where the states are latent. Some studies have also explored the application of sparsity constraints on the estimated latent representations [46]. And it is also extensively discussed in the nonlinear ICA literature [33, 19], in which such a sparsity constraint was added to the mixing function.

### S1.4.3 Mechanism Variability

The assumption of mechanism variability requires that causal dynamics differ between domains, which requires at least one discrete edge variation within the causal transition graphs. This assumption is typically considered reasonable in practical contexts; humans identify distinctions between domains only when the differences are substantial, which often involves the introduction of a new mechanism or the elimination of an existing one. Specifically, this assumption requires a minimal alteration, a single edge change in the causal graph, to be considered satisfied. Consequently, as long as there are significant differences in the causal dynamics among domains, this criterion is fulfilled.

### S1.4.4 Mechanism Function Variability

In this section, we will further discuss the mechanism function variability introduced in Corollary 1. One might question the necessity of this assumption. To illustrate this issue, we claim that if we only assume that the mechanism functions differ across domains but without this extended version of the variability assumption, i.e., for $u \neq u'$, $\mathbf{m}_u \neq \mathbf{m}_{u'}$, then under this proposed framework, the domain variables $u_t$ are generally unidentifiable.

In Figure S4, we present a simple example of the space of $\mathbf{z}_{t+1}$ given a fixed $\mathbf{z}_t$. For the sake of brevity, assume that there are two domains. By the mechanism separability assumption i, the space $\mathcal{Z}$

of $\mathbf{z}_{t+1}$ is divided into two distinct parts, each corresponding to one domain. In this illustration, $\mathcal{C}$ denotes the partition created by the ground truth transition function:

$$\mathbf{z}_{t+1} = \begin{cases} \mathbf{m}_1(\mathbf{z}_t, \epsilon) & \text{if } u_{t+1} = 1, \\ \mathbf{m}_2(\mathbf{z}_t, \epsilon) & \text{if } u_{t+1} = 2. \end{cases} \tag{29}$$

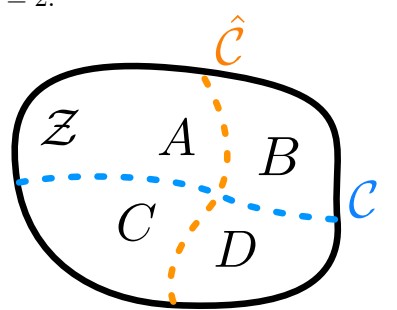

Then the question arises: when the domain assignment is incorrect, that is, $\hat{\mathcal{C}} \neq \mathcal{C}$, can we still get the same observational distributions, or equivalently, can we obtain the same distribution for $\mathbf{z}_{t+1}$?

The answer is yes. For the ground truth transition, $\mathbf{m}_1(\mathbf{z}_t, \epsilon) \in A \cup C$ and $\mathbf{m}_2(\mathbf{z}_t, \epsilon) \in B \cup D$. In the case of an incorrect partition $\hat{\mathcal{C}}$, it is sufficient to have $\hat{\mathbf{m}}_1(\mathbf{z}_t, \epsilon) \in A \cup B$ and $\hat{\mathbf{m}}_2(\mathbf{z}_t, \epsilon) \in C \cup D$. Ensuring that the conditional distribution $p(\mathbf{z}_{t+1} \mid \mathbf{z}_t)$ is matched everywhere, we can create two different partitions on domains, yet still obtain exactly the same observations. That makes the domain variables $u_t$ unidentifiable in the general case.

Figure S4: Correct domain separation $\mathcal{C}$ and incorrect domain separation $\hat{\mathcal{C}}$ of $\mathbf{z}_{t+1}$, given a fixed $\mathbf{z}_t$.

**How does the previous mechanism variability assumption work?** In the assumption of mechanism variability (Assumption iii), the support matrices of the Jacobian of transitions across different domains differ. Consider a scenario where the ground truth partition is $\mathcal{C}$, denoted by $A, C \mid B, D$. If an incorrect estimation occurs, where our estimated partition is $\hat{\mathcal{C}}$, represented as $A, B \mid C, D$, then the estimated transition in domain one should cover the transitions in both $A$ and $B$, and similarly for the second domain. This leads to an increase in complexity within the estimated Jacobian support matrix, as discussed in the previous sections. Consequently, this complexity forces the sets $B$ and $C$ to be empty, resulting in $\hat{\mathcal{C}}$ converging to $\mathcal{C}$.

**How about mechanism function variability?** Roughly speaking, and as demonstrated in our experiments, the mechanism variability assumption previously discussed is already sufficient to identify domain changes in both synthetic and real-world settings. This sufficiency arises because the assumption only requires a single differing spot, even though some transition functions behind some edges may persist across different domains. As long as there is one edge spot that can separate the two domains, this condition is met. In the relatively rare case where all edges in the causal dynamic transition graphs are identical across two different domains and only the underlying functions differ, we can still demonstrate identifiability in this scenario by examining differences in the support of the higher-order partial derivative matrices.

### S1.4.5 Weakly Diverse Lossy Transition

The weakly diverse lossy transition assumption requires that each variable in the latent space can potentially influence a set of subsequent latent variables, and such transformations are typically non-invertible. This implies that given the value of $\mathbf{z}_{t+1}$, it is generally challenging to precisely recover the previous $\mathbf{z}_t$; equivalently, this mapping is not injective. Although this assumption requires some explanation, it is actually considered mild in practice. Often in real-world scenarios, different current states may lead to identical future states, indicating a loss of information. The "weakly diverse" of this assumption suggests that the way information is lost varies between different dimensions, but there is some common part among them, hence the term "weakly diverse". In the visualization example shown in Figure 4, we can clearly see this pattern, in which the scene is relatively simple and it is very likely that in two different frames, the configuration of the scene or the value of the latent variables are the same but their previous states are completely different.

## S2  Experiment Settings

### S2.1  Synthetic Dataset Generation

The synthetic dataset is constructed in accordance with the conditions outlined in Theorems 1 and 2. Transition and mixing functions are synthesized using multilayer perceptrons (MLPs) initialized with

random weights. The mixing functions incorporate the LeakyReLU activation function to ensure invertibility. The dataset features five distinct values for the domain variables, with both the hidden variables $\mathbf{z}_t$ and the observed variables $\mathbf{x}_t$ set to eight dimensions. A total of 1,000 sequences of domain variables were generated. These sequences exhibit high nonstationarity across domains, which cannot be captured with a single Markov chain. This was achieved by initially generating two distinct Markov chains to generate two sequences of domain indices. Subsequently, these sequences were concatenated, along with another sequence sampled from a discrete uniform distribution over the set $\{1, 2, 3, 4, 5\}$, representing the domain indices.

For each sequence of domain variables, we sampled a batch size of 32 sequences of hidden variables $\mathbf{z}_t$ beginning from a randomly initialized initial state $\mathbf{z}_0$. These sequences were generated using the randomly initialized multilayer perceptron (MLP) to model the transitions. Observations $\mathbf{x}_t$ were subsequently generated from $\mathbf{z}_t$ using the mixing function as specified in Eq. 1. Both the transition functions in the hidden space and the mixing functions were shared across the entire dataset. A summary of the statistics for this synthetic dataset is pro-

Table S1: Synthetic Dataset Statistics

| Property | Value |
|---|---|
| Number of State | 5 |
| Dimension of $\mathbf{z}_t$ | 8 |
| Dimension of $\mathbf{x}_t$ | 8 |
| Number of Samples | 32,000 |
| Sequence Length | 15 |

vided in Table S1. For detailed implementation of this data generation process, please refer to our accompanying code in Sec. S3.3.

## S2.2 Real-world Dataset

**Hollywood Extended**  The Hollywood [38] dataset contains 937 video clips with a total of 787,720 frames containing sequences of 16 different daily actions such as walking or sitting from 69 Hollywood movies. On average, each video comprises 5.9 segments, and 60.9% of the frames are background.

**CrossTask**  The CrossTask [39] dataset features videos from 18 primary tasks. According to [40], we use the selected 14 cooking-related tasks, including 2552 videos with 80 action categories. On average, each video in this subset has 14.4 segments, with 74.8% of the frames classified as background.

## S2.3 Mean Correlation Coefficient

MCC, a standard metric in the ICA literature, is utilized to evaluate the recovery of latent factors. This method initially computes the absolute values of the correlation coefficients between each ground truth factor and every estimated latent variable. Depending on the presence of component-wise invertible nonlinearities in the recovered factors, either Pearson's correlation coefficients or Spearman's rank correlation coefficients are employed. The optimal permutation of the factors is determined by solving a linear sum assignment problem on the resultant correlation matrix, which is executed in polynomial time.

## S3 Implementation Details

## S3.1 Prior Likelihood Derivation

Let us start with an illustrative example of stationary latent causal processes consisting of two time-delayed latent variables, i.e., $\mathbf{z}_t = [z_{1,t}, z_{2,t}]$, i.e., $z_{i,t} = m_i(\mathbf{z}_{t-1}, \epsilon_{i,t})$ with mutually independent noises, where we omit the $u_t$ since it is just an index to select the transition function $m_i$. Let us write this latent process as a transformation map $\mathbf{m}$ (note that we overload the notation $m$ for transition functions and for the transformation map):

$$\begin{bmatrix} z_{1,t-1} \\ z_{2,t-1} \\ z_{1,t} \\ z_{2,t} \end{bmatrix} = \mathbf{m} \left( \begin{bmatrix} z_{1,t-1} \\ z_{2,t-1} \\ \epsilon_{1,t} \\ \epsilon_{2,t} \end{bmatrix} \right). \tag{30}$$

By applying the change of variables formula to the map $\mathbf{m}$, we can evaluate the joint distribution of the latent variables $p(z_{1,t-1}, z_{2,t-1}, z_{1,t}, z_{2,t})$ as:

$$p(z_{1,t-1}, z_{2,t-1}, z_{1,t}, z_{2,t}) = p(z_{1,t-1}, z_{2,t-1}, \epsilon_{1,t}, \epsilon_{2,t})/\left|\det \mathbf{J_m}\right|, \tag{31}$$

where $\mathbf{J_m}$ is the Jacobian matrix of the map $\mathbf{m}$, which is naturally a low-triangular matrix:

$$\mathbf{J_m} = \begin{bmatrix} 1 & 0 & 0 & 0 \\ 0 & 1 & 0 & 0 \\ \frac{\partial z_{1,t}}{\partial z_{1,t-1}} & \frac{\partial z_{1,t}}{\partial z_{2,t-1}} & \frac{\partial z_{1,t}}{\partial \epsilon_{1,t}} & 0 \\ \frac{\partial z_{2,t}}{\partial z_{1,t-1}} & \frac{\partial z_{2,t}}{\partial z_{2,t-1}} & 0 & \frac{\partial z_{2,t}}{\partial \epsilon_{2,t}} \end{bmatrix}.$$

Given that this Jacobian is triangular, we can efficiently compute its determinant as $\prod_i \frac{\partial z_{i,t}}{\partial \epsilon_{i,t}}$. Furthermore, because the noise terms are mutually independent, and hence $\epsilon_{i,t} \perp \epsilon_{j,t}$ for $j \neq i$ and $\epsilon_t \perp \mathbf{z}_{t-1}$, we can write the RHS of Eq. 31 as:

$$\begin{aligned}
p(z_{1,t-1}, z_{2,t-1}, z_{1,t}, z_{2,t}) &= p(z_{1,t-1}, z_{2,t-1}) \times p(\epsilon_{1,t}, \epsilon_{2,t})/\left|\det \mathbf{J_m}\right| \quad \text{(because } \epsilon_t \perp \mathbf{z}_{t-1}\text{)} \\
&= p(z_{1,t-1}, z_{2,t-1}) \times \prod_i p(\epsilon_{i,t})/\left|\det \mathbf{J_m}\right| \quad \text{(because } \epsilon_{1,t} \perp \epsilon_{2,t}\text{)}
\end{aligned} \tag{32}$$

Finally, by canceling out the marginals of the lagged latent variables $p(z_{1,t-1}, z_{2,t-1})$ on both sides, we can evaluate the transition prior likelihood as:

$$p(z_{1,t}, z_{2,t} \mid z_{1,t-1}, z_{2,t-1}) = \prod_i p(\epsilon_{i,t})/\left|\det \mathbf{J_m}\right| = \prod_i p(\epsilon_{i,t}) \times \left|\det \mathbf{J_m^{-1}}\right|. \tag{33}$$

Now we generalize this example and derive the prior likelihood below.

Let $\{\hat{m}_i^{-1}\}_{i=1,2,3\dots}$ be a set of learned inverse transition functions that take the estimated latent causal variables, and output the noise terms, i.e., $\hat{\epsilon}_{i,t} = \hat{m}_i^{-1}(u_t, \hat{z}_{i,t}, \hat{\mathbf{z}}_{t-1})$.

Design transformation $\mathbf{A} \to \mathbf{B}$ with low-triangular Jacobian as follows:

$$\underbrace{\left[\hat{\mathbf{z}}_{t-1}, \hat{\mathbf{z}}_t\right]^\top}_{\mathbf{A}} \text{ mapped to } \underbrace{\left[\hat{\mathbf{z}}_{t-1}, \hat{\boldsymbol{\epsilon}}_t\right]^\top}_{\mathbf{B}}, \text{ with } \mathbf{J_{A \to B}} = \begin{pmatrix} \mathbb{I}_n & 0 \\ * & \operatorname{diag}\left(\frac{\partial m_{i,j}^{-1}}{\partial \hat{z}_{jt}}\right) \end{pmatrix}. \tag{34}$$

Similar to Eq. 33, we can obtain the joint distribution of the estimated dynamics subspace as:

$$\log p(\mathbf{A}) = \log p(\hat{\mathbf{z}}_{t-1}) + \underbrace{\sum_{j=1}^n \log p(\hat{\epsilon}_{i,t})}_{\text{Mutually independent noise}} + \log\left(\left|\det(\mathbf{J_{A \to B}})\right|\right). \tag{35}$$

$$\log p(\hat{\mathbf{z}}_t \mid \hat{\mathbf{z}}_{t-1}, u_t) = \sum_{i=1}^n \log p(\hat{\epsilon}_{i,t} \mid u_t) + \sum_{i=1}^n \log\left|\frac{\partial m_i^{-1}}{\partial \hat{z}_{i,t}}\right|. \tag{36}$$

## S3.2 Derivation of ELBO

Then the second part is to maximize the Evidence Lower BOund (ELBO) for the VAE framework, which can be written as:

$$\begin{aligned}
\text{ELBO} &\triangleq \log p_{\text{data}}(\{\mathbf{x}_t\}_{t=1}^T) - D_{KL}(q_\phi(\{\mathbf{z}_t\}_{t=1}^T \mid \{\mathbf{x}_t\}_{t=1}^T) \,\|\, p_{\text{data}}(\{\mathbf{z}_t\}_{t=1}^T \mid \{\mathbf{x}_t\}_{t=1}^T)) \\
&= \mathbb{E}_{\mathbf{z}_t} \log p_{\text{data}}(\{\mathbf{x}_t\}_{t=1}^T \mid \{\mathbf{z}_t\}_{t=1}^T) - D_{KL}(q_\phi(\{\mathbf{z}_t\}_{t=1}^T \mid \{\mathbf{x}_t\}_{t=1}^T) \,\|\, p_{\text{data}}(\{\mathbf{z}_t\}_{t=1}^T \mid \{\mathbf{x}_t\}_{t=1}^T)) \\
&= \mathbb{E}_{\mathbf{z}_t} \log p_{\text{data}}(\{\mathbf{x}_t\}_{t=1}^T \mid \{\mathbf{z}_t\}_{t=1}^T) - \mathbb{E}_{\mathbf{z}_t} \left( \log q_\phi(\{\mathbf{z}_t\}_{t=1}^T \mid \{\mathbf{x}_t\}_{t=1}^T) - \log p_{\text{data}}(\{\mathbf{z}_t\}_{t=1}^T) \right) \\
&= \mathbb{E}_{\mathbf{z}_t} \left( \log p_{\text{data}}(\{\mathbf{x}_t\}_{t=1}^T \mid \{\mathbf{z}_t\}_{t=1}^T) + \log p_{\text{data}}(\{\mathbf{z}_t\}_{t=1}^T) - \log q_\phi(\{\mathbf{z}_t\}_{t=1}^T \mid \{\mathbf{x}_t\}_{t=1}^T) \right) \\
&= \mathbb{E}_{\mathbf{z}_t} \left( \underbrace{\sum_{t=1}^T \log p_{\text{data}}(\mathbf{x}_t \mid \mathbf{z}_t)}_{-\mathcal{L}_{\text{Recon}}} + \underbrace{\sum_{t=1}^T \log p_{\text{data}}(\mathbf{z}_t \mid \mathbf{z}_{t-1}, u_t) - \sum_{t=1}^T \log q_\phi(\mathbf{z}_t \mid \mathbf{x}_t)}_{-\mathcal{L}_{\text{KLD}}} \right)
\end{aligned}$$

$$(37)$$

### S3.3 Reproducibility

All experiments are performed on a GPU server with 128 CPU cores, 1TB memory, and one NVIDIA L40 GPU. Our code is also available via `https://github.com/xiangchensong/ctrlns`. For synthetic experiments, we run the baseline methods with implementation from `https://github.com/weirayao/leap` and `https://github.com/xiangchensong/nctrl`. For real-world experiments, the implementation is based on `https://github.com/isee-laboratory/cvpr24_atba`.

### S3.4 Hyperparameter and Train Details

For synthetic experiments, the models were implemented in `PyTorch 2.2.2`. We trained the VAE network using the AdamW optimizer with a learning rate of $5 \times 10^{-4}$ and a mini-batch size of 64. Each experiment was conducted using three different random seeds and we reported the mean performance along with the standard deviation averaged across these seeds. The coefficient for the $L_2$ penalty term was set to $1 \times 10^{-4}$, which yielded satisfactory performance in our experiments.

the most accurate way to enforce sparsity is through the $L_0$ norm. However, since calculating the gradient for it is challenging, $L_p$ norms are commonly used as approximations. We also tested alternative settings such as $L_1$ penalty or larger coefficients for $L_2$ norm, and we found that the setting we used in this paper ($L_2$ with coefficient $1 \times 10^{-4}$) provided the best stability and performance.

All other hyperparameters of the baseline methods follow their default values from their original implementation. For real-world experiments, we follow the same hyperparameter setting from the baseline ATBA method. In the Hollywood dataset, we used the default 10-fold dataset split setting and calculated the mean and standard derivation from those 10 runs. For the CrossTask dataset, we calculate the mean and standard derivation using five different random seeds.

### S3.5 Enforce Invertibility

In our experimental setup, using reconstruction loss already provides strong identifiability results (MCC>0.95), and it is widely used in the identifiability literature. During our experiments, we found that flow-based methods are usually less efficient and typically take longer to converge. Since our main contribution is to address the challenge of unknown domain variables, this choice is orthogonal to our theoretical contribution. Therefore, we followed the existing work for the design of the estimation for the mixing function. As also mentioned in [47], the reconstruction loss-based framework can definitely be extended to flow-based methods, especially in environments where invertibility is a critical issue and computation is not a top priority in the estimation process.

### S3.6 Encode Domain Variables

As introduced in 4, we utilize $U$ distinct networks to capture the different transitions. One may question whether this choice introduces redundant parameters and suggest employing parameter-sharing networks, as in [22, 45]. However, our decision to use separate transition networks is based on the assumptions regarding the complexity of the transition functions, which we regularize through the sparsity of the Jacobian matrix. Implementing parameter-sharing techniques would significantly complicate the optimization process, as updating parameters for one domain would immediately alter the Jacobian matrix for another domain.

Despite using individual transition networks, they remain lightweight compared to the entire framework. Even in synthetic scenarios where the framework is relatively small, each transition network accounts for only approximately 2.3% of the total parameters. This proportion becomes even smaller in real-world applications with larger encoder-decoder frameworks.

# S4 Extended Related Work

## S4.1 Causal Discovery with Latent Variables

Various studies have focused on uncovering causally related latent variables. For example, [48–50] use vanishing Tetrad conditions [51] or rank constraints to detect latent variables in linear-Gaussian models, whereas [52–55] rely on non-Gaussianity in their analyses of linear, non-Gaussian models. Additionally, some methods seek to identify structures beyond latent variables, leading to hierarchical structures. Certain hierarchical model-based approaches assume tree-like configurations, as seen in [56–59], while other methods consider a more general hierarchical structure [55, 50]. Nonetheless, these approaches are restricted to linear frameworks and encounter increasing difficulties with complex datasets, such as videos.

## S4.2 Causal Temporal Representation Learning

In the context of sequence or time series data, recent advances in nonlinear Independent Component Analysis (ICA) have leveraged temporal structures and nonstationarities to achieve identifiability. Time-contrastive learning (TCL) [10] exploits variability in variance across data segments under the assumption of independent sources. Permutation-based contrastive learning (PCL) [11] discriminates between true and permuted sources using contrastive loss, achieving identifiability under the uniformly dependent assumption. The i-VAE [13] uses Variational Autoencoders to approximate the joint distribution over observed and nonstationary regimes. Additionally, (i)-CITRIS [60, 61] utilizes intervention target information to identify latent causal factors. Other approaches such as LEAP [22] and TDRL [23] leverage nonstationarities from noise and transitions to establish identifiability. GCIM [62] builds on the theoretical insights from LEAP by implementing a clustering algorithm to address the challenge of unobserved domain indices. However, it does not achieve identifiability of the domain variables. CaRiNG [63] extended TDRL to handle non-invertible generation processes by assuming sequence-wise recoverability of the latent variables from observations.

All the aforementioned methods either assume stationary fixed temporal causal relations or that the domain variables controlling the nonstationary transitions are observed. To address unknown or unobserved domain variables, HMNLICA [15] integrates nonlinear ICA with a hidden Markov model to automatically model nonstationarity. However, this method does not account for the autoregressive latent transitions between latent variables over time. IDEA [26] combines HMNLICA and TDRL by categorizing the latent factors into domain-variant and domain-invariant groups. For the variant variables, IDEA adopts the same Markov chain model as HMNLICA, while for the invariant variables, it reduces the model to a stationary case handled by TDRL. Both iMSM [24] and NCTRL [25] extend this Markov structure approach by incorporating transitions in the latent space but continue to assume that the domain variables follow a Markov chain.

Moreover, causal temporal representation learning has been explored in the reinforcement learning literature [64, 45]. These works primarily focus on the relationship between states and actions, typically relying on direct state observations. In contrast, our setting involves recovering meaningful latent causal variables from observational data.

## S4.3 Weakly-supervised Action Segmentation

Weakly-supervised action segmentation techniques focus on dividing a video into distinct action segments using training videos annotated solely by transcripts [38, 65, 43, 66, 42, 67, 41, 68–74]. Although these methods have varying optimization objectives, many employ pseudo-segmentation for training by aligning video sequences with transcripts through techniques like Connectionist Temporal Classification (CTC) [65], Viterbi [41, 43, 42, 70, 71, 73], or Dynamic Time Warping (DTW) [67, 69]. For instance, [65] extends CTC to consider visual similarities between frames while evaluating valid alignments between videos and transcripts. Drawing inspiration from speech recognition, [71, 41, 73] utilize the Hidden Markov Model (HMM) to link videos and actions. [66] initially produces uniform segmentations and iteratively refines boundaries by inserting repeated actions into the transcript. [43] introduces an alignment objective based on explicit context and length models, solvable via Viterbi, to generate pseudo labels for training a frame-wise classifier. Similarly, [42] and [70] propose novel learning objectives but still rely on Viterbi for optimal pseudo segmentation. Both [67, 69] use DTW to align videos to both ground-truth and negative transcripts, emphasizing the contrast between them. However, except for [66], these methods require frame-by-frame calculations, making them inefficient. More recently, alignment-free methods have been introduced to enhance efficiency. [68] learns from the mutual consistency between frame-wise classification and category/length pairs of a segmentation. [44] enforces the output order of actions to match the transcript order using a novel loss function. Although POC [44] is primarily set-supervised, it can be extended to transcript supervision, making its results relevant for comparison.

