# OpenReview forum: "Causal Temporal Representation Learning with Nonstationary Sparse Transition"
_NeurIPS.cc/2024/Conference — NeurIPS 2024 poster_

### Official Review · Reviewer_6hAo · 2024-06-20

**Soundness:** 3
**Presentation:** 3
**Contribution:** 3
**Rating:** 6
**Confidence:** 2

**Summary:**

Most existing causal temporal representation learning models either assume the domain variables to be observable, or have a Markov prior over them. This paper first comes up an identifiability theory for sequential data affected by non-stationary latent causal processes under unknown distributional shifts. In particular, a novel causal temporal representation learning method is developed to identify the latent causal variables with time-delayed causal relations. Moreover, without relying on prior knowledge on distributional shifts, the domain variables can be inferred from sequential observations. The final experiments on both synthetic and real-world action segmentation data, demonstrate the superior performance of the novel method in estimating domain variables and latent causal processes.

**Strengths:**

Originality:
It is the first attempt to develop causal temporal representational learning to identify both the domain variables and the latent causal processes from sequential observations, without relying on distributional or structural knowledge.

Quality:
The theorectical aspects of the paper look very sound, and absolutely pass the acceptance bar of the NeurIPS.

Clarity:
Overall, the paper is well written. Both the target problems and the main techniques are well presented!

Significance:
The developed identifiability theory for domain variables and latent causal processes from sequential observations under unknown distributional or structural prior knowledge, may have impacts for advancing the relevant causal representation learning methods.

**Weaknesses:**

Throughout the paper, I did not find the details about how to set up the parameters of the neural nets for the novel CTRL framework, which could be crucial for the others to pick up the work.

**Questions:**

Except the action segmentation tasks, can the authors provide more real applications including healthcare and finance, as mentioned in the boarder impacts.

**Limitations:**

Yes, the paper discusses the potential negative societal impact of the work.

---

> ### Author Rebuttal · Authors · 2024-08-03
>
> We sincerely thank the reviewer for acknowledging our work and contribution, and we also thank you for providing valuable questions and suggestions. Please kindly find our response below:
>
> > W1. Parameter Settings
>
> We appreciate the reviewer for raising this suggestion, which has improved our readability. In terms of the neural network design, our CtrlNS model is based on the baseline implementation of NCTRL. Specifically, we replaced the HMM module with our sparse transition module while keeping the encoder and decoder modules consistent with the baseline. The sparse transition module is implemented using MLPs to create gating functions in conjunction with the transition function employed in our methods. More details can be found in the codebase provided in Appendix S3.3. In light of the suggestion, we have included such details in the experiments section of the updated version.
>
> > Q1. Additional Real Applications (Healthcare and Finance)
>
> We highly appreciate your valuable suggestion about applying our method to better verify its effectiveness. Beyond the action segmentation task, our proposed method can also be used in sensory data in the healthcare domain. For example, Apple Watch sensor data can be utilized to automatically detect health condition changes and remind the user to seek medical help if the domain variable indicates a risky health condition. A similar design is also useful in the finance domain when monitoring market data to perform early detection of black swan events.

---

> ### Author Response · Authors · 2024-08-13
>
> Dear Reviewer 6hAo
>
> We sincerely appreciate the time and effort you dedicated to reviewing our submission and providing such insightful comments. Your feedback is invaluable to us. If there are any unresolved concerns or additional thoughts, we would be more than happy to address them.
>
> Thank you again for your thoughtful contributions.
>
> Best regards,
> Authors of submission 4804

---

### Official Review · Reviewer_1QG3 · 2024-07-05

**Soundness:** 3
**Presentation:** 3
**Contribution:** 2
**Rating:** 5
**Confidence:** 4

**Summary:**

The paper focuses on causal temporal representation learning for non-stationary time series. It adopts a sparse transition assumption, aligned with intuitive human understanding, and presents identifiability results from a theoretical perspective. Based on the theoretical result, the authors introduce a novel framework, Causal Temporal Representation Learning with Nonstationary Sparse Transition (CtrlNS), designed to leverage the constraints on transition sparsity and conditional independence to reliably identify both distribution shifts and latent factors. Experiments on synthetic and real-world datasets demonstrate the effectiveness of the proposed method in recovering latent variables and domain indices.

**Strengths:**

1.The paper focuses on the issue of causal discovery in nonstationary time series, which is a fascinating and significant area of research.

2.The authors claim to have proposed a method that establishes identifiability of nonstationary nonlinear ICA for general sequence data without prior knowledge of domain variables, which is an extremely challenging problem.

3.The authors have conducted experiments on both synthetic and real-world datasets, validating the effectiveness of the model, and have appropriately discussed the limitations of the model.

**Weaknesses:**

1.The authors lack discussion of some existing work, for example, the paper [1] also demonstrates identifiability by clustering nonstationary spatiotemporal data into different domains without prior knowledge. It is essential for authors to further discuss the similarities and differences between the two papers and alleviate my concerns about the novelty of this work.

2.In lines 259-261 of this paper, the authors mention ensuring the invertibility of the modeled mixing function through reconstruction loss, which is insufficient. Invertible Neural Networks (INNs) should be employed to ensure invertibility, as there will inevitably be errors in the reconstruction process.

[1] Zhao, Yu, et al. "Generative Causal Interpretation Model for Spatio-Temporal Representation Learning." Proceedings of the 29th ACM SIGKDD Conference on Knowledge Discovery and Data Mining. 2023.

**Questions:**

Please see the Weaknesses.

**Limitations:**

The authors have appropriately discussed the limitations and societal impacts of the model.

---

> ### Author Rebuttal · Authors · 2024-08-04
>
> We thank the reviewer for providing valuable questions and suggestions. Please kindly find our response below:
>
> > Related Work: GCIM [1]
>
> We appreciate the reviewer for highlighting the additional related work, GCIM. We will include this work in the related work section. Regarding the comparison, both works explore finding latent causal representations from observational data only and handle the nonstationary setting with unobserved domain variables. However, there are several differences between our work and GCIM:
>
> * First, from a theoretical aspect, GCIM only shows the identifiability of latent variables $\mathbf{z}$ under the condition that the domain variables $u$ are observed. In the estimation part, as described in Section 3.2 of GCIM, it uses a Domain Adapter to empirically estimate domain variables, but the identifiability of domain variables is not addressed in their proof. One of our main contributions is to provide the identifiability result for the domain variables, clearly differentiating our work from GCIM.
>
> * Second, regarding the problem setting, as stated in Eq. 1 of [1], GCIM assumes the same causal structure of the transition function but with different nonstationary noise terms in different domains, which is similar to the setting in LEAP [2]. In contrast, our setting involves i.i.d. noise terms with different transition functions for each domain.
>
> > Invertible Neural Networks
>
> We sincerely thank the reviewer for their valuable feedback and insightful suggestions. We fully agree that invertible neural networks, such as flow-based methods, are theoretically more aligned with our approach. In our experiments, we adopted the reconstruction loss-based method for the following reasons:
>
> * In our experimental setup, using reconstruction loss already provides strong identifiability results (MCC>0.95), and it is widely used in the identifiability literature, such as in [3,4].
> * During our experiments, we found that flow-based methods are usually less efficient and typically take longer to converge. Since our main contribution is to address the challenge of unknown domain variables, this choice is orthogonal to our theoretical contribution. Therefore, we followed the existing work for the design of the estimation for the mixing function.
> * As also mentioned in [3], the reconstruction loss-based framework can definitely be extended to flow-based methods, especially in environments where invertibility is a critical issue and computation is not a top priority in the estimation process.
>
> We have included this discussion in the updated version of our paper and once again thank the reviewer for highlighting this important point.
>
> [2] Yao, Weiran, et al. "Learning Temporally Causal Latent Processes from General Temporal Data." International Conference on Learning Representations, 2022. openreview.net/forum?id=RDlLMjLJXdq.
>
> [3] Zhang, Kun, et al. "Causal Representation Learning from Multiple Distributions: A General Setting." 2024. arXiv, https://arxiv.org/abs/2402.05052.
>
> [4] Song, Xiangchen, et al. "Temporally Disentangled Representation Learning under Unknown Nonstationarity." NeurIPS, 2023.

---

> ### Author Response · Authors · 2024-08-13
>
> Dear Reviewer 1QG3
>
> We sincerely appreciate the time and effort you dedicated to reviewing our submission and providing such insightful comments. Your feedback is invaluable to us. If there are any unresolved concerns or additional thoughts, we would be more than happy to address them.
>
> Thank you again for your thoughtful contributions.
>
> Best regards,
> Authors of submission 4804

---

### Official Review · Reviewer_Zo2n · 2024-07-11

**Soundness:** 2
**Presentation:** 2
**Contribution:** 1
**Rating:** 4
**Confidence:** 2

**Summary:**

The paper focuses on temporally causal representation learning, where the goal is to recover a latent causal process from nonstationary observation sequences. Existing works represent the source of the nonstationarity by either known domain variables or (autocorrelated) unknown domain variables with Markov structure. The paper proposes a sparsity based constraint on the latent transitions to achieve the identifiability of the domain variables. Coupled with the identified domain variable, they make use of the sufficient variability assumption to identify the latent causal process. Based on this theoretical framework, they introduce a sequential VAE based model, which encourages the independent noise variables and the sparse transitions. They evaluate (i) the performance on the latent process identifiability on a synthetic dataset following their model definition, and (ii) the usefulness of their model on a weakly-supervised video action segmentation task on real-world datasets of daily actions and cooking.

**Strengths:**

* Compared to the existing work, the paper tackles the identifiability of the domain variables from a new perspective: the domain variable's effect on the transition functions. This perspective is in line with seeing distribution shifts as interventions to the underlying system, and could be valuable in some related tasks such as learning causal relations by taking actions.
* I appreciate the effort for the real-world experiment, though it only evaluates the prediction of the domain variable.

**Weaknesses:**

* As is common in identifiability literature, the assumptions of the theoretical framework do not seem to be testable on a real-world setup.
* Definition 6 (Weakly diverse lossy transitions) seems unrealistic and is not clear from the text: (i) The motivation for the “Lossy” is not clear to me, how is there a causal link between two variables, while changing the causal parent does not affect the effect variable? (ii) On the clarity of the definition: perhaps, some graphical model examples could help here. In addition, the assumption is not discussed in the main text, except stating in lines 227-228 that it is “a mild and realistic condition in the real-world scenarios, allowing for identical future latent states with differing past states”. Why is it mild and realistic?
* It is not clear to me why having domain variables without a Markov structure is a better model of the real-world environments than domain variables with Markov structure. I don’t see why environments in the real-world would swap at random multiple times along a sequence of time points. For example, in the provided real-world experiment, ground-truth actions in the video in Figure S5 occur in blocks and they are autocorrelated. The weak-supervision of the action order used in this experiment also supports their autocorrelation.
* According to the point above, the synthetic experimental setup seems unrealistic as it has randomly sampled domain variables.
* I think the first contribution statement in Lines 59-61 is ambiguous: "this is the first identifiability result that handles nonstationary time-delayed causally related latent temporal processes without prior knowledge of the domain variables". What is meant by "without the prior knowledge of domain variables" is a bit vague, while the difference with the existing work [Song+23] seems to be the assumptions on the temporal dependence of the domain variables. [Song+23] assumes temporally-dependent domain variables and predicts the unknown domain variables with an HMM, while this work assumes temporally independent domain variables and predicts them from sequential observations.

[Song+23]. Xiangchen Song, Weiran Yao, Yewen Fan, Xinshuai Dong, Guangyi Chen, Juan Carlos Niebles, Eric Xing, and Kun Zhang. "Temporally disentangled representation learning under unknown nonstationarity" Neurips, 2023.

**Questions:**

* Some assumptions are motivated in lines 219-224 as being aligned with human intuition: “separability states that if human observers cannot distinguish between two domains, it is unlikely that automated systems can achieve this distinction either. Secondly, variability requires that the transitions across domains are significant enough to be noticeable by humans, implying that there must be at least one altered edge in the causal graph across the domains.” I find it hard to follow this reasoning. Why are these the cases?
* For both experiments, architectural details are missing. Besides, some information on the synthetic experimental setups is missing: what are the functions $\mathbf{m}$ and $\mathbf{g}$ used?, What is the train/val/test split?, What parts of the experiment change with the changing seed?
* It is not clear to me what the model predicts on the action segmentation task. I think it predicts the action class per frame given the video frames and the action order. If so, why do you report MoF, IoU and IoD, instead of accuracy? How do you provide the weak-supervision information to the model? Why is the baseline NCTRL excluded from this task? Its generative model seems to fit the task well.
* It is not clear what the sparsity loss is. In line 266, it is said to be the L2 norm of the parameter in the transition estimation functions. Which parameter is this?

**Limitations:**

* The main limitation, the assumptions being untestable for a real-world setup, is not discussed in the main text.

---

> ### Author Rebuttal · Authors · 2024-08-05
>
> We thank the reviewer for providing valuable questions and suggestions. Please kindly find our response below:
>
> > W1. Non-Testable Assumptions in Real-World Setup
>
> Thank you for bringing attention to this common issue in the identifiability literature. We appreciate your insight. With your permission, we would like to add this statement to the paper to make this aspect more explicit.
>
> > W2. Assumption of Weakly Diverse Lossy Transition
>
> * To address the reviewer's question about the causal link: if there is a causal link $A \to B$, it is possible to have cases where changing the value of $A$ doesn't change the distribution of $p(B|A)$. For example, $B = Relu(A) + \epsilon_B$, we can clearly see that changing the value of $A$ within the interval $(-\inf, 0)$ doesn't change the distribution of $B|A$. i.e., as long as there exists $a_1 \neq a_2$ such that $p(B|A=a_1) \neq p(B|A=a_2)$, then the causal link $A \to B$ exists.
> * We thank the reviewer for the suggestion to further clarify the definition. We will add more illustrations and examples like the one above to make it clearer. However, since this is an assumption on the quantitative inference process, it may not be faithfully reflected in the graphical models.
> * Due to page limitations, we defer the discussion of the feasibility issue of this assumption to Appendix S1.4.5. In light of your comment, we feel it is better to avoid using terms like "mild" and "realistic" to prevent potentially overclaiming our contribution.
>
> > W3. Why Non-Markovian is Better? Real-World Autocorrelation
>
> From our understanding, the autocorrelation mentioned by the reviewer here refers to temporally dependent processes. However, it may not be reasonable to further assume that the states can be modeled by a single Markov chain. There are many situations where the Markov assumption is too restrictive. For example, by concatenating a state sequence generated from a Markov chain with a sequence generated from a temporally independent distribution, this simple scenario can already break HMM-based models (details discussed in the synthetic experiment lines 328-332), let alone more complex real-world cases.
>
> > W4. Synthetic Experiment Setup + W5. Assumptions on the Temporal Dependence of the Domain Variables
>
> We would like to clarify that our method doesn't assume temporally independent domain variables; on the contrary, we allow a mix of temporally dependent and independent situations. As described in Appendix S2.1, in the synthetic setup the domain variables are not independently randomly sampled. Instead, it concatenates a state sequence generated from a Markov chain with a state sequence generated from a temporally independent distribution. Such a mixed setup is more complex compared with temporally dependent only, and the synthetic experiment also supports our claim.
>
> > W5. Without the Prior Knowledge + Q1. Aligned with Human Intuition
>
> We defer the response to the global response as other reviewers also mentioned similar questions.
>
> > Q2. Some Missing Details for Experiments
>
> * For synthetic experiments, encoder $\mathbf{g}^{-1}$ and decoder $\mathbf{g}$ are implemented with a VAE structure, and transition networks $\mathbf{m}_u$ are implemented with MLPs with a gating function to select the domain values.
> * Following existing work [Song+23], we randomly select 10% of the data for testing and leave the rest for training.
> * We use three random seeds for the training process, influencing the network parameter initialization and the optimization process.
> * For real-world experiments, we follow the settings and model architectures in the ATBA [Xu+24] baseline and report the testing results based on their existing dataset split, which is widely used in the action segmentation community. The results with random seeds also follow the convention in that domain, and details are provided in Appendix S3.4.
> * Please kindly let us know if our explanation properly addresses your concerns. By the way, all design details of the network and architecture can be found in the code (Appendix S3.3).
>
> > Q3. Metrics in Action Segmentation Task
>
> Yes, the output of the model is the action class per frame in the video. As described in lines 359 and 391, the Mean-over-Frames (MoF) denotes the mean accuracy of the framewise action prediction. Following existing literature in action segmentation, we also report IoU and IoD.
>
> > Q3. How Weak Supervision is Provided
>
> We follow the baseline method ATBA in using weak supervision. This information is provided as predefined ordered action patterns (so-called transcripts), and these transcripts are used in the boundary alignment process by lowering the score for candidate boundary assignments that violate the transcript order. We refer the reviewer to Sec 3.4 of [Xu+24] for detailed design.
>
> > Q3. Why NCTRL is Not in the Action Segmentation Task
>
> We noticed there are already HMM-based methods in the baselines, and we have lines 374-394 directly discussing this line of methods. However, we agree that adding NCTRL for comparison makes the argument more convincing. The updated results in CrossTask can be found in the table below. We can see that since the Markov assumption may not be fullfilled by the transition of the action in real-world setting, the NCTRL cannot solve this problem well.
>
> |Method|MoF|IoU|IoD|
> |---|---|---|---|
> |NCTRL|50.7|12.9|23.3|
> |CtrlNS|**54.0**|**15.7**|**23.6**|
>
> > Q4. Sparsity Loss
>
> The parameters of transition estimation functions refer to the neural networks we used to estimate the transition functions $\mathbf{m}_{u}$. The sparse loss, as indicated in Eq. 15, is approximated using the $L_2$ norm of the parameters of those neural networks.
>
> [Xu+24] Xu. Angchi, et al. "Efficient and Effective Weakly-Supervised Action Segmentation via Action-Transition-Aware Boundary Alignment." CVPR 2024.

---

> ### Comment · Reviewer_Zo2n · 2024-08-12
>
> Dear authors,
>
> Thank you for taking time to answer my concerns, however, the following points are still unclear for me:
> * For W2,
>   * As far as I understand, the example provided, $B = Relu(A) + \epsilon_B$, does not satisfy the lossy assumption as clearly $\frac{\partial m_B}{\partial A} \neq 0$ everywhere. It is just $\frac{\partial m_B}{\partial A} = 0$ for $A \in (- \inf, 0)$. When $\frac{\partial m_B}{\partial A} = 0$ for all values of $A$, I still do not see how it can be a causal parent.
> * For W3, your real-world dataset shows that *a sequence generated from a temporally independent distribution* is unrealistic. A realistic setup would be to test your model on only state sequences generated from a Markov chain.
> * For Q2,
>   * architecture details are still missing,
>   * it would be better to see the datasets generated from different seeds,
>   * the details on ground-truth transition and mixing function details are still missing,
> * For Q4, the sparsity loss is wrong. L1 norm is commonly used to encourage sparsity. As this is the only architectural novelty, I am not sure how much the second contribution, (2) the CtrlNS framework, makes sense.

---

> ### Author Response · Authors · 2024-08-13
>
> Dear Reviewer Zo2n,
>
> Thank you for providing your feedback. Please see our further response below:
>
> > As far as I understand, the example provided, $B=Relu(A) + \epsilon_{B}$, does not satisfy the lossy assumption as clearly $\frac{\partial m_{b}}{\partial A} \neq 0$ everywhere. It is just $\frac{\partial m_{b}}{\partial A} = 0$ for $A\in (-inf,0)$. When $\frac{\partial m_{b}}{\partial A} = 0$ for all values of $A$, I still do not see how it can be a causal parent.
>
> We would like to highlight the definition of the lossy assumption in line 136:
>
> - Line 136: **There exists** an open set $S_{t,i,j}$ such that changing $z_{t−1,i}$ within this set will not change the value of $m_j$.
>
> As we only require **there exists** a subset of $\mathcal{Z}$ where the partial derivative is zero, it indicates that we don't require "$\frac{\partial m_{b}}{\partial A} = 0$ for all values of $A$", hence this should not influence the existence of the causal link.
>
> > For W3, your real-world dataset shows that a sequence generated from a temporally independent distribution is unrealistic. A realistic setup would be to test your model on only state sequences generated from a Markov chain.
>
> We would like to note that:
>
> 1. The state sequence used in our experiment is not purely generated from a temporally independent distribution, and our method doesn't require the state sequence to be generated from a temporally independent distribution.
> 2. Our method can allow a mixture of independent and dependent settings (i.e., we can handle either dependent, independent, or both). Our synthetic experiment setup specifically explores this mixed case. It is clearly temporally dependent and can not be modeled by a Markov chain.
>
> We respectfully disagree with the statement "using only state sequences generated from a Markov chain is a realistic setup" in our synthetic experiment for the following reasons:
>
> 1. Synthetic experiments are designed to validate the proposed theory and clearly distinguish it from existing baselines. Since we claim that our method achieves domain variable identifiability even when the data does not originate from a single Markov chain, it is essential that our synthetic experiments directly test this scenario. Specifically, in comparison to NCTRL, we demonstrate that NCTRL is unable to handle the more general case where $u_t$ is not generated from a Markov chain, whereas our method is capable of managing such complexity.
>
> 2. Regarding the setting mentioned by the reviewer, "only state sequences generated from a Markov chain," we understand that this is the synthetic experiment setup used in the NCTRL paper. We do not see the necessity for this setup in our synthetic experiments, as both NCTRL and our method are capable of handling this scenario and we are not claiming to outperform NCTRL in this scenario.
>
> 3. From the real-world experiments comparing NCTRL and our method, it is evident that our method outperforms NCTRL. This suggests that assuming the state sequence is generated from a Markov chain is not suitable for real-world settings.
>
>
> > For Q2, architecture details are still missing.
>
> We kindly ask the reviewer to specify which aspects of the architecture described in Section 4.1 remain unclear. We are more than happy to provide additional details and clarifications as needed.
>
> > It would be better to see the datasets generated from different seeds.
>
> We thank the reviewer for suggesting this additional experiment setting. We are in the process of expanding the experiments to include more datasets generated from multiple seeds. However, given it is close to the discussion deadline, we will do our utmost to include the results before the discussion concludes.
>
> > The details on ground-truth transition and mixing function details are still missing.
>
> The transition and mixing functions are implemented as randomly initialized MLPs with hidden dim of 32 with input and output dim 8. All implementation details can be found in our codebase provided in Appendix S3.3. Hope the attached code can make it clear.
>
> > For Q4, the sparsity loss is wrong. L1 norm is commonly used to encourage sparsity. As this is the only architectural novelty, I am not sure how much the second contribution, (2) the CtrlNS framework, makes sense.
>
> We would like to clarify that the most accurate way to enforce sparsity is through the $L_0$ norm. However, since calculating the gradient for $L_0$ is challenging, $L_p$ norms are commonly used as approximations. As indicated in line 1083 of Appendix S3.4, we tested both $L_1$ and $L_2$ norms and both work well in optimal case (MCC: 0.9690 vs 0.9704), and the setting we employed in the paper was selected for its superior stability.
>
> Regarding the contribution, we have already conducted an ablation study in the real-world experiment section. The results show that removing the sparsity loss leads to a performance drop, which supports the effectiveness of our proposed method.

---

> ### Author Response · Authors · 2024-08-13
> **Additional experiment result on multiple random seeds.**
>
> > It would be better to see the datasets generated from different seeds.
>
> We include additional experimental results using three different random seeds to generate synthetic datasets. We compare our method with NCTRL and report the mean and standard deviation of the results across those three datasets in the following table. It can be observed that since NCTRL cannot generalize beyond the Markov assumption, its performance is weaker than our method and is also less stable in comparison.
>
> | Method | MCC($z_t$)|Acc($u_t$)|
> |--------|------|------|
> |NCTRL|50.99$\pm$3.77|57.51$\pm$13.23|
> |CtrlNS|95.04$\pm$1.39|97.09$\pm$0.89|
>
> We sincerely appreciate the time and effort you dedicated to reviewing our submission and providing such insightful comments. If there are any unresolved concerns or additional thoughts, we would be more than happy to address them.
>
> Best regards,
>
> Authors of submission 4804

---

### Official Review · Reviewer_3MKp · 2024-07-22

**Soundness:** 3
**Presentation:** 3
**Contribution:** 3
**Rating:** 7
**Confidence:** 3

**Summary:**

The paper introduces CtrlNS, a causal temporal representation learning framework based on a sparse transition assumption to identify distribution shifts without strong prior knowledge of the domain variables. Theoretical and experimental results show CtrlNS effectively identifies distribution shifts and latent factors, outperforming existing methods.

**Strengths:**

- Causal temporal representation learning under nonstationarity is an important problem. The paper replaces somewhat unrealistic and strong assumptions on the domain variable made by the prior works.
- Theoretical analysis provides a rigorous foundation for the proposed framework. While I did not check the proofs in detail, the proof flows are convincing, and the results are intuitive and reasonable.
- Another strength of the paper is its evaluation on realistic task setup (i.e., weakly supervised action segmentation). This illustrates the practical applicability of the proposed method.
- Finally, the paper is well-written and well-organized.

**Weaknesses:**

The authors argue that the paper derives identifiability results *“without prior knowledge of the domain variables”* (e.g., line 46, 61). However, this is an overstatement since CtrlNS still requires some prior knowledge or assumptions, such as the ground-truth number of different domains $U$. Moreover, while I understand it does not rely on Markov assumption of prior works (e.g., NCTRL), the proposed framework still requires a different set of assumptions such as weakly diverse lossy transition and mechanism sparsity. In fact, I am not sure whether the authors’ assumption can be claimed as “weaker” compared to prior works. Therefore, I feel those statements need to be toned down, and more discussion on the comparisons of those assumptions is suggested.

**Questions:**

- It seems that the framework requires the prior knowledge of the number of environments $U$. This assumption should be made more explicit. Also, what happens under the misspecification of $U$?
- For the mathematical rigor, the statements need to take measure-zero sets into account. For example, I think the conditions $p(S_{t, i, j})>0$ (line 136), $p(S_{t, i})>0$ and $p(S_{t, i, j}\setminus S_{t, i})>0$ (line 141) should be included, just to name a few.
- The framework requires $U$ different transition networks, which could be a potential weakness, especially in terms of scalability. The discussion on this point would further strengthen the paper. For example, an efficient parameter sharing of networks [3] may alleviate such issues.
- The proposed work assumes no contemporaneous causal relationships. (I know this is a common assumption in this line of work) How can this assumption be relaxed? With the interventional data, is it possible for the proposed framework to generalize to the existence of instantaneous effects, maybe similar to iCITIRS [1]?
- A number of works in RL setting also consider the identifiability under the non-stationary process [2, 3]. For example, [3] considers the identifiability of both domain variables and causal structures. I suggest the authors include these related works in different areas.
- The utilization of mechanism sparsity for identifiability is also explored in [4]. How is it different from the sparse transition assumption in CtrlNS?

(minor)

- Style file seems to be different from the official NeurIPS template (e.g., fonts).

***References***

[1] Lippe, Phillip, et al. "Causal Representation Learning for Instantaneous and Temporal Effects in Interactive Systems." *The Eleventh International Conference on Learning Representations*.

[2] Feng, Fan, et al. "Factored adaptation for non-stationary reinforcement learning." *Advances in Neural Information Processing Systems* 35 (2022): 31957-31971.

[3] Hwang, Inwoo, et al. "Fine-Grained Causal Dynamics Learning with Quantization for Improving Robustness in Reinforcement Learning." *International Conference on Machine Learning*. PMLR, 2024.

[4] Xu, Danru, et al. "A Sparsity Principle for Partially Observable Causal Representation Learning." *International Conference on Machine Learning*. PMLR, 2024.

**Limitations:**

.

---

> ### Author Rebuttal · Authors · 2024-08-05
>
> We sincerely thank the reviewer for acknowledging our work and contribution, and we also thank the reviewer for providing valuable questions and suggestions. Please kindly find our response below:
>
> > W1+Q1. Claim on Prior Knowledge
>
> We appreciate the reviewer for their valuable suggestions to make our presentation more precise. We have revised this claim to a more moderate statement: "doesn't rely on knowledge of the prior distribution of the domain variables." We have explicitly included the requirement of a predefined number of $U$ in the updated paper.
>
> > W2. Assumption Comparison with NCTRL
>
> We completely agree that there is no strict winner or loser between our assumptions and those of NCTRL. We have weaker assumptions on the distribution of domain variables but place additional assumptions on transition functions. We intend for our method to serve as an alternative solution when the distribution of domain variables is unknown.
>
> > Q1. What happens under misspecification of $U$?
>
> Specifying an insufficient number will definitely hurt the identifiability, causing multiple domains to become entangled. However, as discussed in Appendix S1.1.1 (lines 709 to 711), as long as the allocated number of $U$ is greater than or equal to the ground truth, identifiability can still be achieved. This also suggests a potential future work direction on how to further relax this requirement and automatically find the number of $U$, starting from a sufficiently large number and gradually decreasing to a suitable value.
>
> > Q2+Q(minor). More Precise Mathematical Statements and Style File Issue
>
> We thank the reviewer for pointing this out. We have added those conditions and carefully proofread our statements to ensure their precision when updating the draft. Regarding the style file issue, we have identified that the problem stems from the `lmodern` package, which we have removed in the updated version.
>
> > Q3. Why Separate Transition Networks
>
> We thank the reviewer for raising this question and completely agree with the reviewer's concern. The reason we use different transition networks is due to our assumptions on the complexity of the transition functions, where we regularize using the sparsity of the Jacobian matrix. Using parameter-sharing tricks would make the optimization problem very difficult, since in this setting, updating the parameter for one domain will immediately change the Jacobian matrix for another domain. Despite using separate transition networks, they are lightweight compared with the whole framework; even in a synthetic setting where the whole framework is relatively small, each transition network only takes ~2.3% of the parameters. This percentage is even lower in real-world cases with larger encoder-decoder framework. We have included a discussion of parameter-sharing networks in our updated version.
>
> > Q4. Contemporaneous Relation
>
> We noticed that there is concurrent work [5] that also takes advantage of the sparsity of the transition to establish identifiability with instantaneous relations. Their setting is stationary, but we believe our work can be utilized to further extend [5] to a nonstationary setting.
>
> > Q4. Interventional Data
>
> We thank the reviewer for such a valuable suggestion. We believe that leveraging interventional data can further relax the assumptions made in this work. For example, the one-edge difference among different domains may not be necessary with interventional data, as we can always compose paired data to separate the domains. We leave further discussion in this line as a future work direction.
>
> > Q5. Related Work in RL Community
>
> We thank the reviewer for reminding us to discuss related works including [2,3] in RL. RL focuses more on the relation between states and actions, while we focus more on finding meaningful representation from observational data. For instance, [3] uses direct observation of states in RL environments, while our setting requires recovering meaningful latent causal variables from observational data. We have included more discussions within this line of research in the updated version.
>
> > Q6. Difference from [4]
>
> Thanks for pointing to this paper. We noticed that [4] also utilized sparsity constraints to establish identifiability results. However, the way sparsity constraints are applied is different from our method. The sparsity in [4] was placed on the values of latent variables $\mathbf{Z}$, as there is no temporal process in their setting. In our case, we place the sparsity constraint on the transition of the latent variables, allowing the values of latent variables in our method to be almost arbitrarily dense.
>
> [5] Li, Zijian, et al. "On the Identification of Temporally Causal Representation with Instantaneous Dependence." 2024, arXiv, https://arxiv.org/abs/2405.15325.

---

> ### Author Response · Authors · 2024-08-14
>
> Dear Reviewer 3MKp,
>
> We would like to express our sincere gratitude for the time and effort you invested in reviewing our submission. We greatly appreciate your acknowledgment of our contribution and the insightful comments you provided. We have carefully considered your feedback and have made the response, which we hope address your concerns. Should you have any further comments or additional suggestions, we would be more than happy to discuss them.
>
> Thank you once again for your valuable input.
>
> Best regards,
>
> Authors of submission 4804

---

### Author Rebuttal · Authors · 2024-08-05

We thank all reviewers for providing valuable and insightful questions and suggestions on our work. We found that our claim of "without prior knowledge of domain variables" and the "motivation to align with human intuition" are mentioned by multiple reviewers. We give a comprehensive response here.

> Regarding Without Prior Knowledge of Domain Variables

We would like to thank reviewer Zo2n for summarizing that NCTRL assumes temporally-dependent domain variables and predicts the unknown domain variables with an HMM. Such an assumption on the distribution form of the domain variables is widely used in the literature when dealing with unobserved domain variables.

In contrast, our claim is that we do not need such information about the domain variables. In our case, the domain variables can be either temporally dependent or independent; it doesn't matter. As long as the transitions controlled by these domain variables are sufficiently different, we can still recover those domain variables from observational data.

We thank both reviewers 3MKp and Zo2n for mentioning this aspect, and we have revised this claim to a more moderate statement: "doesn't rely on knowledge of the prior distribution of the domain variables" to avoid overclaiming our contribution. We hope this helps to clarify the message we want to convey to the audience.

> Regarding the Motivation of the Assumptions Aligning with Human Intuition

We thank reviewer Zo2n for raising this question and thank reviewer 3MKp for mentioning the comparison of assumptions with existing work. We leverage this opportunity to clarify and emphasize the central idea behind our work.

* We assume human observers perform the reasoning process with meaningful representation in their minds. Given that humans have this ability, our goal is to enable machines to also possess this capability. Under the scope of this work, we are exploring the decision boundary on the conditions under which a machine can act like a human to identify the domains from observational data.
* We aim to quantify the significance of such changes among different domains. Our assumption of at least one edge being different across domains is just a sufficient condition in which the changes are significant enough to establish identifiability for machines.
* It is not a necessary condition; there are certainly cases where all edges are the same for two domains, but humans can still tell the difference.
* We partially explored this case in Sec 3.2 remark, where we use higher-order sparsity to achieve identifiability. We leave the exploration of the rest of the cases for future work and also encourage the community to join us in further exploring this decision boundary between machine perception and human perception.

---

### Decision · Program_Chairs · 2024-09-25

**Decision:**

Accept (poster)

**Comment:**

The paper presents a causal temporal representation learning framework for identifying distribution shifts and latent factors in nonstationary time series without strong prior knowledge of domain variables. It introduces a sparse transition assumption and provides theoretical identifiability results for both domain variables and latent causal processes from sequential observations.

Strengths:

+ Tackles an important problem of causal temporal representation learning under nonstationarity with sparse transition assumptions than prior work

+ Provides identifiability results

+ Evaluates the method on both synthetic and realistic tasks (action segmentation)

Weaknesses:

+ Some key assumptions (e.g. weakly diverse lossy transitions) can be unrealistic

+ Lacks comparison to relevant related work (that may rely on different identification assumptions)